# A new mouse model of Charcot-Marie-Tooth 2J neuropathy replicates human axonopathy and suggest alteration in axo-glia communication

Ghjuvan'Ghjacumu Shackleford[1,2,3]*, Leandro N. Marziali[1,2], Yo Sasaki[4], Anke Claessens[3], Cinzia Ferri[3], Nadav I. Weinstock[1,2], Alexander M. Rossor[5], Nicholas J. Silvestri[1,2], Emma R. Wilson[1,2], Edward Hurley[1,2], Grahame J. Kidd[6], Senthilvelan Manohar[7], Dalian Ding[7], Richard J. Salvi[7], M. Laura Feltri[1,2☯]*, Maurizio D'Antonio[3☯], Lawrence Wrabetz[1,2☯]

1 Department of Neurology, Institute for Myelin and Glia Exploration, Jacobs School of Medicine and Biomedical Sciences, State University of New York at Buffalo, Buffalo, New York, United States of America, 2 Department of Biochemistry, Institute for Myelin and Glia Exploration, Department Biochemistry and Neurology, Jacobs School of Medicine and Biomedical Sciences, State University of New York at Buffalo, New York, United States of America, 3 Biology of Myelin Unit, Division of Genetics and Cell Biology, San Raffaele Scientific Institute, Milan, Italy, 4 Needleman Center for Neurometabolism and Axonal Therapeutics and Department of Genetics, Washington University School of Medicine in Saint Louis, St. Louis, Missouri, United States of America, 5 Department of Neuromuscular Diseases, UCL Queen Square Institute of Neurology, London, United Kingdom, 6 Department of Neurosciences, Lerner Research Institute, Cleveland Clinic, Cleveland, Ohio, United States of America, 7 Center for Hearing and Deafness, State University of New York at Buffalo, Buffalo, New York, United States of America

☯ These authors contributed equally to this work.
* shackleford.ghjuvanghjacumu@hsr.it (GGS); mlfeltri@buffalo.edu (MLF)

**Data Availability Statement:** All relevant data are within the manuscript and its Supporting Information files.

## Abstract

Myelin is essential for rapid nerve impulse propagation and axon protection. Accordingly, defects in myelination or myelin maintenance lead to secondary axonal damage and subsequent degeneration. Studies utilizing genetic (CNPase-, MAG-, and PLP-null mice) and naturally occurring neuropathy models suggest that myelinating glia also support axons independently from myelin. Myelin protein zero (MPZ or P0), which is expressed only by Schwann cells, is critical for myelin formation and maintenance in the peripheral nervous system. Many mutations in *MPZ* are associated with demyelinating neuropathies (Charcot-Marie-Tooth disease type 1B [CMT1B]). Surprisingly, the substitution of threonine by methionine at position 124 of P0 (P0T124M) causes axonal neuropathy (CMT2J) with little to no myelin damage. This disease provides an excellent paradigm to understand how myelinating glia support axons independently from myelin. To study this, we generated targeted knock-in $Mpz^{T124M}$ mutant mice, a genetically authentic model of T124M-CMT2J neuropathy. Similar to patients, these mice develop axonopathy between 2 and 12 months of age, characterized by impaired motor performance, normal nerve conduction velocities but reduced compound motor action potential amplitudes, and axonal damage with only minor compact myelin modifications. Mechanistically, we detected metabolic changes that could lead to axonal degeneration, and prominent alterations in non-compact myelin domains such as paranodes, Schmidt-Lanterman incisures, and gap junctions, implicated in

**Funding:** We are grateful for funding from: Charcot-Marie-Tooth association (https://www.cmtausa.org) to L.W and G.G.S. and Fondazione Telethon (https://www.telethon.it/en) to M.D (GGP19099) The funders had no role in study design, data collection and analysis, decision to publish, or preparation of the manuscript.

**Competing interests:** The authors have declared that no competing interests exist.

Schwann cell-axon communication and axonal metabolic support. Finally, we document perturbed mitochondrial size and distribution along $Mpz^{T124M}$ axons suggesting altered axonal transport. Our data suggest that Schwann cells in P0T124M mutant mice cannot provide axons with sufficient trophic support, leading to reduced ATP biosynthesis and axonopathy. In conclusion, the $Mpz^{T124M}$ mouse model faithfully reproduces the human neuropathy and represents a unique tool for identifying the molecular basis for glial support of axons.

## Author summary

Charcot-Marie-Tooth (CMT) neuropathies are a large family of incurable peripheral nerve disorders. Despite extensive clinical and genetic heterogeneity, axonal degeneration is the common end point of all the type of CMT. Thus, a major question is why axons degenerate and how to protect them. Over the years, it has become clear that myelinating glial cells, which are in close contact with axons, are essential for axonal support. However how glial cells support axons remains only partially understood. Here, we generated and characterized an animal model of Charcot-Marie-Tooth 2J (CMT2J), an axonal inherited neuropathy due to mutation in the Myelin Protein Zero (*Mpz*) gene. *Mpz* is expressed only in Schwann cells, the peripheral myelin-forming glia, but not in axons, making this the model unique to contribute to our understanding on how glia support axons. Our model reproduces very closely most aspects of the human neuropathy and reveals several alterations in domains crucial for axoglial communication. We also detected metabolic abnormalities in peripheral nerves of these mice that are known to be associated with axonal degeneration. Our work sheds light on the cellular and molecular mechanisms of axoglial communication and axonal degeneration, with implications for a plethora of neurodegenerative diseases.

## Introduction

In the peripheral nervous system (PNS), Schwann cells (SC) make myelin by wrapping their plasma membranes around axons. Myelin is a multilamellar lipid-rich structure containing a set of proteins (in the PNS: mostly myelin protein zero [MPZ, P0], peripheral myelin protein 22 [PMP22], and myelin basic protein [MBP]) essential for its compaction. Among these structural proteins, P0, encoded by the *MPZ* gene, is the most abundant, accounting for 45% of the total protein expressed in PNS myelin [1]. P0 is a single-pass transmembrane glycoprotein with an immunoglobulin-like fold in its extracellular domain that is expressed exclusively by SC. Crystallography of P0 and X-ray diffraction analysis of myelin suggest that P0 forms homotetramers interacting *in trans* to hold together adjacent wraps of myelin membrane [2]. As evidence of the crucial role of P0 in myelination and myelin compaction [3], *Mpz* deficient mice produce few myelin layers that are poorly compacted [4].

Charcot-Marie-Tooth (CMT) neuropathies are the most common inherited neurological disorders, affecting 1/2500 people [5]. CMT neuropathies are caused by alterations in over 80 genes encompassing approximately 1,000 independent mutations [6]. Based on nerve conduction velocity (NCV), CMT are mainly classified as demyelinating CMT type 1 (CMT1) (NCV<38 m/s) or axonal CMT type 2 (CMT2) (NCV>38 m/s) hereditary neuropathies. CMT1B, represents a special challenge because it is caused by more than 200 diverse mutations in *MPZ*, resulting in different toxic gain of function mechanisms and various related clinical

phenotypes [7,8]. Initially, the majority of P0 mutations identified were associated with demyelinating neuropathies. CMT1B patients typically present with early onset disease with extremely slow NCV (<10–20 m/s). Surprisingly, since P0 is only expressed in SC but not in neurons, several mutations in P0 [9–12], such as the substitution of threonine by methionine at position 124 (P0T124M), were shown to cause an axonal neuropathy referenced as CMT2J. After around 40 years of age, patients with T124M-CMT2J begin experiencing symptoms, such as lancinating pain and fast-progressive weakness of the lower limbs. NCVs vary widely among T124M-CMT2J patients and can be normal, slightly reduced similar to that seen in CMT2, or as slow as in CMT1 but never as low as in CMT1B. However, the amplitudes of motor evoked potentials are strikingly reduced in patients carrying the P0T124M mutation. Sural nerve biopsy samples exhibit regenerative clusters and a marked reduction of myelinated axons, but with little to no myelin damage. Moreover, T124M-CMT2J patients exhibit early signs of sensory abnormalities, such as pupillary abnormalities and hearing loss [13–15]. Beside its myelin producing function, these observations suggest either that P0 in SC contributes to axonal support or that SC expressing P0T124M produce deleterious signals contributing to axonal degeneration.

Axonal degeneration is a common endpoint of peripheral neuropathies [16] and understanding the glial processes that contribute to axon protection and degeneration is considered the key to cure neuropathies. Yet, this topic is poorly understood. Axonal degeneration is uncoupled from demyelination in CMT2J, providing a unique opportunity to understand how myelinating glia support axons independently of myelin. We generated an authentic mouse model of CMT2J carrying the P0T124M mutation, representing, to the best of our knowledge, the first animal model for CMT2J disease and the first model of human axonal neuropathy caused by a mutation in a protein of compact myelin. Our findings show that the $Mpz^{T124M}$ mouse model closely recapitulates the axonopathy and clinical aspects observed in CMT2J patients. Alterations in non-compact myelin, metabolic changes, and mitochondria dysfunction in $Mpz^{T124M}$ mutants suggest that axonal loss is the result of a defect in SC-to-axon communication. Our results highlight potential mechanisms of how a mutation in P0 causes axonal degeneration without demyelination and underlie the fundamental role of SC in axonal support.

## Results

### $Mpz^{T124M}$ mice display progressive motor defects

Using homologous recombination, we engineered a genetically authentic T124M-CMT2J mouse model (**S1 Fig**). Like CMT2J patients, the mice harboring the P0T124M mutation (referred to as TM mice in figures) exhibit signs of late-onset, progressive peripheral neuropathy. Starting at 6 months of age, and more obviously at 12 months, we noticed limb clasping behavior, clawed hind paws, and Achilles' tendon retraction (**Fig 1A and 1B**). These neurological abnormalities were not fully penetrant, reflecting the heterogeneity of symptoms observed in T124M-CMT2J patients, and were more prominent in $Mpz^{T124M}$ homozygous ($Mpz^{T124M/T124M}$) mice than in heterozygous ($Mpz^{T124M/+}$) mice.

We tested the motor performance of $Mpz^{T124M}$ mice in the rotarod and beam walking tests and found no differences from wild-type (WT) mice at 2 months of age (**Fig 1C**). However, at 6 months of age (**Fig 1D**), we noticed a trend towards a reduced motor capacity in $Mpz^{T124M}$ mice. By 12 months (**Fig 1E**), $Mpz^{T124M}$ mice remained on the accelerating rotarod half as long as WT mice. Results from the beam walking test further demonstrated motor impairments (**Fig 1F and S1–S3 Movies**). At 6 months of age, $Mpz^{T124M}$ mice exhibited more foot slips (**Fig 1F and 1G**), and $Mpz^{T124M/T124M}$ mice were 25% slower than the WT mice in

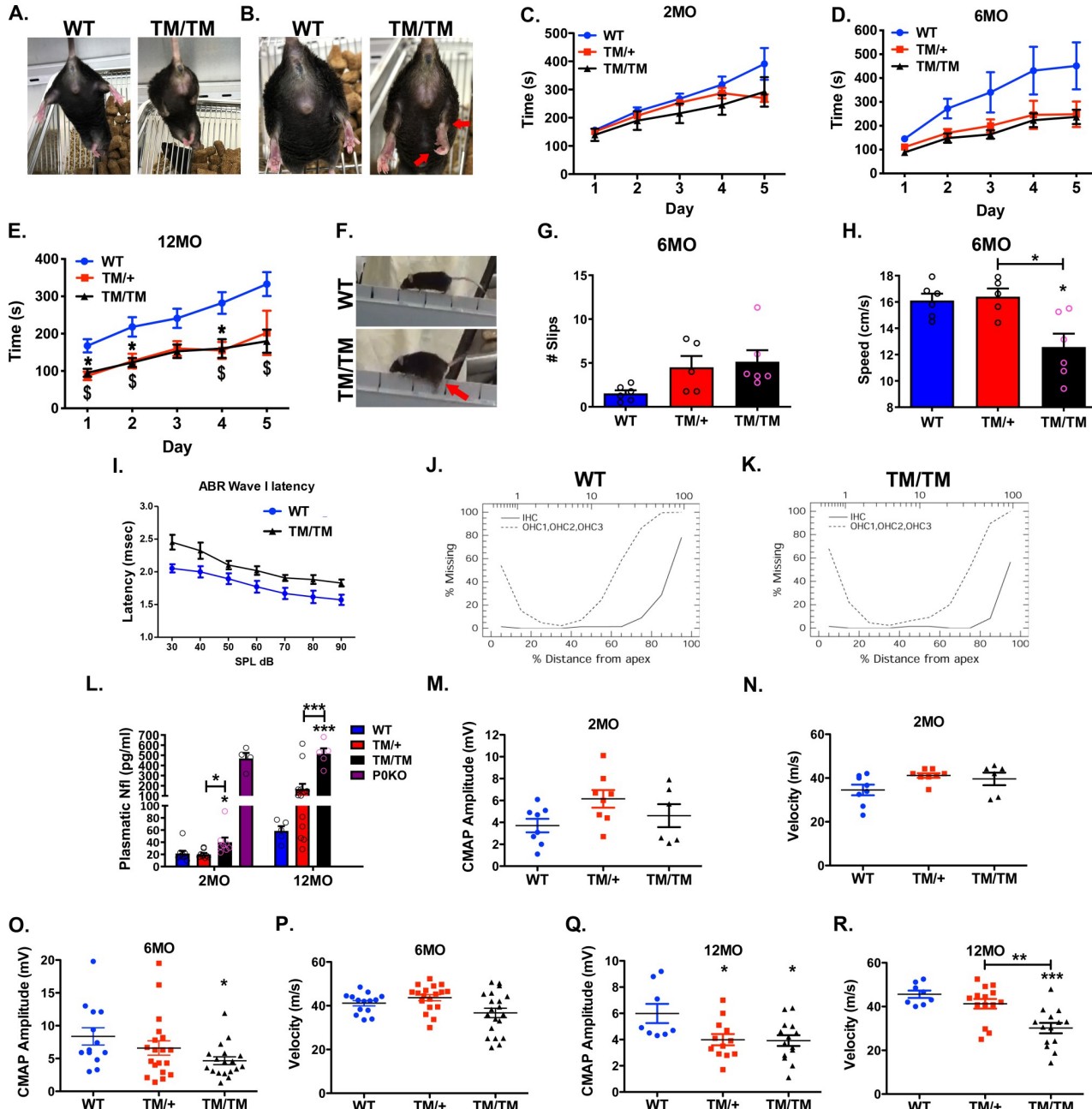

**Fig 1. Clinical impairments in *Mpz*^T124M mice.** (**A and B**) External signs of neuropathy in 12-month-old *Mpz*^T124M/T124M (TM/TM) mice. (**A**) Representative picture of clasping behavior. (**B**) Clawed hind paws and Achilles' tendon retraction are indicated by arrows. (**C to G**) Locomotion is impaired in *Mpz*^T124M mice. Accelerating rotarod test at 2 (**C**), 6 (**D**), and 12 (**E**) months of age. No differences were observed at 2 months of age. Compared to that of wild-type (WT) mice, the performance of *Mpz*^T124M mice was worse at 6 months of age and significantly altered at 12. *n* (animals) ≥ 4 per genotype. Two-way ANOVA [2 month old: Time: F (1.840, 38.64) = 30.54; P<0.0001, Genotype F (2, 21) = 2.177; P = 0.1382; 6 month old: Time: F (1.723, 43.06) = 19.55, P<0.0001; Genotype: F (2, 25) = 4.412; P = 0.0228; 12 month old: Time: F (1.291, 23.24) = 11.51; P = 0.0013; Genotype: F (2, 18) = 4.590; P = 0.0245] with Tuckey's *post hoc*. * *p < 0.05*: significant difference between WT and *Mpz*^T124M/+ (TM/+); ^$*p < 0.05*: significant difference between WT and *Mpz*^T124M/T124M. (**F to H**) Beam walking test at 6 months of age. (**F**) Representative pictures of beam walking test for WT and *Mpz*^T124M/T124M mice. Arrow indicates a slip. Quantification of slips (**G**) and speed (**H**). *n* (animals) ≥ 5 per genotype. One-way ANOVA [Slip: F (2, 14) = 3.430; P = 0.0613; Speed: F (2, 14) = 7.839; P = 0.0052]. (**I to K**) Auditory neuropathy in *Mpz*^T124M mice. (**I**) ABR wave I latency measurements in WT and *Mpz*^T124M/T124M mice at 11 months of age. Cochleograms of WT (**J**) and *Mpz*^T124M/T124M (**K**) mice at 12 months of age. *n* (animals) ≥ 4 per genotype. (**L**) Plasmatic neurofilament light (Nfl) concentrations in WT, *Mpz*^T124M/+, and *Mpz*^T124M/T124M mice at 2 and 12 months of age. Plasma from 4-month-old P0 null mice was used as a positive control. *n* (animals) ≥ 5 per genotype. One-way ANOVA [2 month old: F (2, 23) = 4.857; P = 0.0174; 12 month old: F (2, 21) = 14.83; P<0.0001]. (**M to R**) Electrophysiological analysis. Amplitudes of compound muscle action potentials (CMAPs) and nerve conduction velocities (NCVs) were measured at 2 (**M and N**), 6 (**O and P**), and 12 (**Q and R**) months of age. *n* (nerves)

$\geq$ 6 per genotype. One-way ANOVA [2 month old: amplitude: $F_{(2, 19)} = 2.53$; $P = 0.1058$, NCV: $F_{(2, 19)} = 2.83$; $P = 0.0837$; 6 month old: amplitude: $F_{(2, 47)} = 3.142$; $P = 0.05$; NCV: $F_{(2, 49)} = 4.394$; $P = 0.0176$; 12 month old: amplitude: $F_{(2, 31)} = 4.608$; $P = 0.0177$; NCV: $F_{(2, 31)} = 10.95$; $P = 0.0003$]. $^*p < 0.05$, $^{**}p < 0.01$, $^{***}p < 0.001$ by Tukey's *post hoc* tests (**G to R**) after one-way ANOVA. Graphs indicate means ± SEMs.

crossing the rod (**Fig 1H**). Thus, $Mpz^{T124M}$ mice develop a peripheral neuropathy and display locomotion impairment.

## Hearing loss in $Mpz^{T124M}$ mice

Hearing loss is one of the most consistently observed symptoms in T124M-CMT2J patients [13,14]. To determine if hearing loss also occurs in our mouse model, we used click stimuli to obtain auditory brainstem evoked potential recordings (ABR). The ABR trace consists of five major peaks or waves occurring in the first 5–7 ms following stimulus onset. Wave I, which has a latency of approximately 1.8 ms at high intensities, reflects neural activity of the auditory nerve and subsequent peaks represent to a first approximation synchronized neural responses from successfully more proximal regions in the auditory brainstem (i.e. cochlear nuclei, superior olive, lateral lemniscus, and inferior colliculus). Because only the distal part of the auditory nerve is myelinated by SC, while more central structures in the CNS are myelinated by oligodendrocytes, we focused our attention on ABR wave I. By 11 months of age, the click-evoked latency of ABR wave I was markedly increased at all stimulus intensities in $Mpz^{T124M/T124M}$ mice compared to WT mice (**Fig 1I**). The increase in latency is suggestive of a neural conduction delay in the auditory nerve similar to that observed in T124M-CMT2J patients. To determine if this latency prolongation was caused by loss of sensory hair cells, we performed a cochleogram assay to assess the percentage loss of outer hair cells (OHC) and inner hair cells (IHC) from the base to the apex of the cochlea. Mean cochleogram revealed losses of OHC and to a lesser extent IHC in the basal half of the cochlea of both $Mpz^{T124M/T124M}$ and WT mice. Because there were no significant differences in the magnitude of OHC and IHC lesions between WT and $Mpz^{T124M/T124M}$ mice, the longer wave I latencies in $Mpz^{T124M/T124M}$ mice compared to WT mice are presumably caused by defect on peripheral auditory nerve fibers (**Fig 1J and 1K**). Abnormal ABRs are characteristic of the auditory neuropathy observed in late-onset CMT1B patients harboring the P0Y145S mutation [17], patients with *PMP22* mutations [18,19], and those with *GJB1* (gap junction beta 1) mutations [18,20], suggesting that hearing impairment in those with the P0T124M mutation may be due in part to degeneration of the distal part of the auditory nerve.

## Level of plasmatic Neurofilament light is increased in $Mpz^{T124M}$ mice

Plasmatic neurofilament light (pNfl) is emerging as a biomarker for a variety of neurological diseases and positively correlates with the severity of peripheral neuropathies [21]. Remarkably, pNfl concentrations were increased 2-fold in $Mpz^{T124M/T124M}$ mice at 2 months of age and 8-fold at 12 months of age. In $Mpz^{T124M/+}$ mice, pNfl concentrations were trending toward an increase (3-fold) at 12 months of age (**Fig 1L**). These data suggest that pNfl level could also serve as a biomarker in T124M-CMT2J patients and may correlate with disease progression.

## Electrophysiological alterations in $Mpz^{T124M}$ mice

Electrophysiological analyses showed no differences between $Mpz^{T124M}$ and WT mice at 2 months of age (**Fig 1M and 1N**). At 6 months of age, there was a trend toward reduced compound muscle action potential (CMAP) amplitude in $Mpz^{T124M/+}$ mice and a significantly reduced CMAP amplitude in $Mpz^{T124M/T124M}$ mice (**Fig 1O**) but normal NCVs (**Fig 1P**). At 12

months, both $Mpz^{T124M/T124M}$ and $Mpz^{T124M/+}$ mice exhibited significantly reduced CMAP amplitudes (**Fig 1Q**), consistent with the observed neuromuscular impairment, but only $Mpz^{T124M/T124M}$ mice had significantly lower NCVs (**Fig 1R**). Altogether, these results show that the P0T124M mutation causes a peripheral neuropathy with negative functional impact on axons before myelin.

## $Mpz^{T124M}$ mice develop a progressive axonopathy with subtle myelin defects

We next studied the consequence of P0T124M mutation on nerve morphology (**Figs 2 and S2**). Because P0 is the most aboundant myelin protein and because P0 mutations are responsible for hypomyelinating, dysmyelinating, and demyelinating CMT1B neuropathies, we first studied the ultrastructure of myelin sheaths in sciatic nerves from $Mpz^{T124M}$ mice (**Fig 2A–2D**). At 2 and 6 months of age, there were no significant differences in myelin thickness between WT and $Mpz^{T124M}$ mice. By contrast, we detected thicker myelin sheaths, reflected by significantly lower g-ratios, in 12- and 18-month-old $Mpz^{T124M/T124M}$ and $Mpz^{T124M/+}$ mice (**Fig 2E**). The increase in myelin thickness was predominantly observed for small-diameter axons (**Figs 2F and S3**). To determine whether the increased thickness is attributable to a defect in myelin compaction, we measured myelin periodicity. There was a slight but not significant, increase of myelin periodicity in $Mpz^{T124M}$ mice (**Fig 2G and 2H**). Consistent with this result, the expression of major myelin proteins (myelin-associated glycoprotein [MAG], PMP22, 2',3'-cyclic nucleotide 3'-phosphodiesterase and [CNPase]) was not affected in $Mpz^{T124M}$ sciatic nerve at 2 and 6 months of age. Similar results were obtained at 18 months of age, except for a decrease in CNPase expression (**S4 Fig**). Moreover, the mRNA levels of master transcription factors controlling myelination and SC differentiation (*sex determining region Y-box 2* [*Sox2*], *inhibitor of DNA binding 2* [*Id2*] [22], *Jun proto-oncogene* [*c-Jun*] [23], *Pou domain class 3 transcription factor 1* [*Oct6*] [24], *early growth response 2* [*Krox20*] [25]) were not altered in $MPZ^{T124M}$ sciatic nerve at 2 and 12 months of age (**S5 Fig**). These results corroborate the observations of samples from T124M-CMT2J patients, in which myelination process and compact myelin was not or only mildly altered and was well organized [14,15].

Although compact myelin ultrastructure was not markedly altered by the P0T124M mutation, we noticed, comparable to that in T124M-CMT2J patients [14], occasional supernumerary SC processes and onion bulbs (**Fig 2A, 2B and 2I**), suggesting some active de-/remyelination in $Mpz^{T124M/T124M}$ mice. In line with the normal expression of immature and differentiation SC markers as *c-Jun* and Sox2, onion bulbs were rare (0.1–0.2% of myelinated fibers) and did not progress over time (**Fig 2J**) as would be expected in pure demyelinating neuropathy models [26,27]. Finally, in $Mpz^{T124M/T124M}$ mice at least 6 months of age, we detected myelin balloons (**Figs 2K and S2D**), which are swellings thought to be formed by fluid influx inside the intraperiod line of the myelin sheath. Pressure created from fluid influx compresses the axon against one side of the balloon and could be responsible for axonal degeneration [28].

Although we only noticed subtle changes in the myelin sheath *per se*, the most striking phenotype observed in $Mpz^{T124M}$ nerves was related to fibers degeneration (arrowheads **Figs 2A–2D and S2**). A morphometric analysis revealed a significant reduction of myelinated fibers in sciatic nerves from $Mpz^{T124M/T124M}$ mice at 12 and 18 months of age (**Fig 3A**). At the latter timepoint, we noticed a slight shift toward smaller axon diameters in $Mpz^{T124M/T124M}$ mice, suggesting potential loss of large-diameter axons (**Fig 3B**). Starting at 6 months of age for $Mpz^{T124M/T124M}$ mice and 18 months of age for $Mpz^{T124M/+}$ mice, we observed a massive loss of fibers within digital nerves (in the toes), suggesting a predominant distal dying back axonal

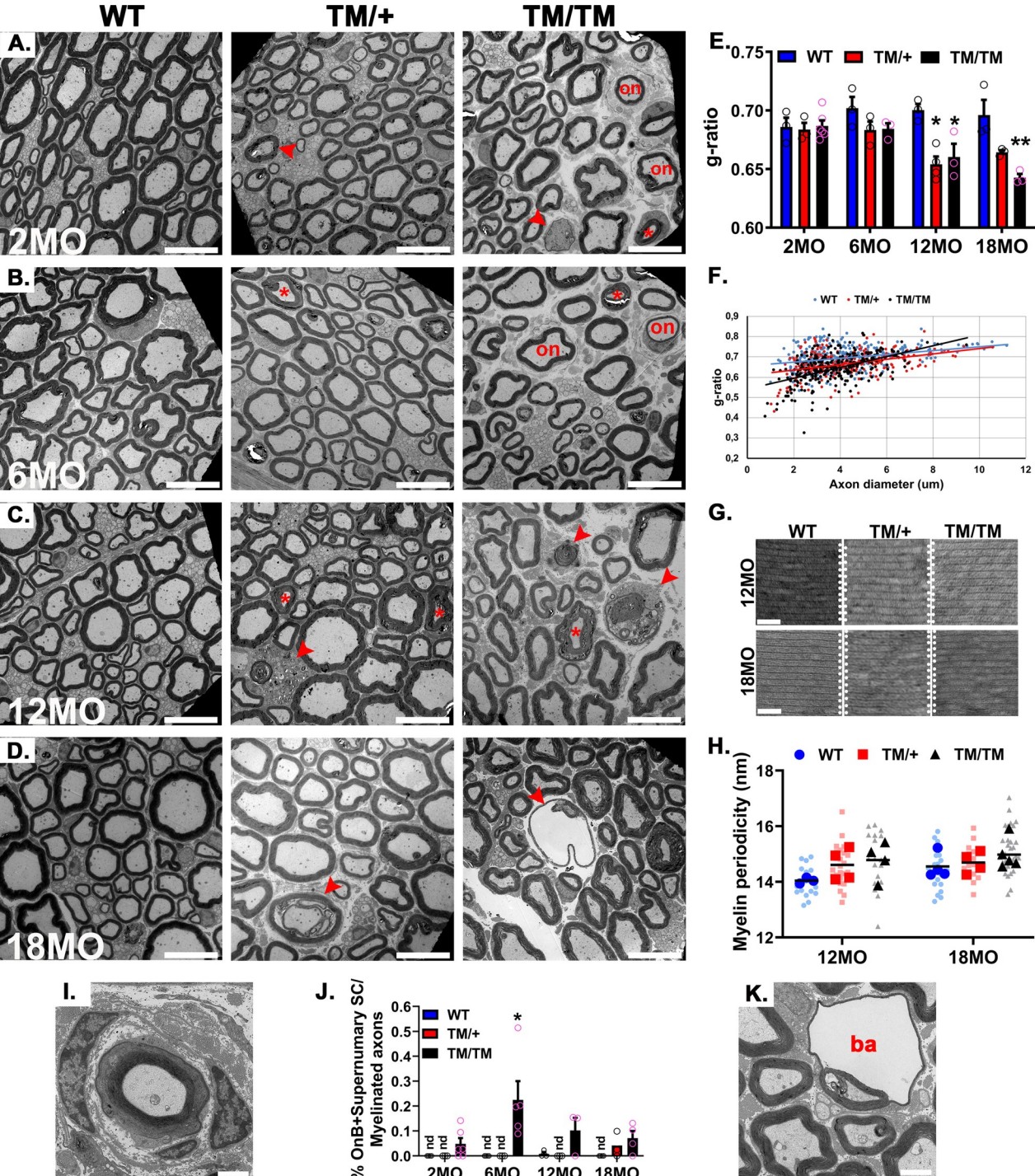

**Fig 2. Morphological analysis of $Mpz^{T124M}$ nerves.** (**A** to **D**) Representative electron micrographs of wild-type (WT), $Mpz^{T124M/+}$ (TM/+), and $Mpz^{T124M/T124M}$ (TM/TM) sciatic nerve cross sections at 2 (**A**), 6 (**B**), 12 (**C**), and 18 (**D**) months of age. Arrowheads indicate degenerative figures, asterisks indicate SLI, and "on" indicate onion bulbs and supernumerary SC. Scale bars: 10 μm. (**E**) g-ratio analysis shows hypermyelination of $Mpz^{T124M}$ fibers starting at 12 months of age. One-way ANOVA [2 month old: F (2, 9) = 0.0738, p = 0.9294; 6 month old: F (2, 6) = 1.994, p = 0.2168; 12 month old: F (2, 7) = 9.655, p = 0.0097; 18 month old: F (2, 6) = 11.8, p = 0.0083]. (**F**) g-ratio as a function of axonal diameter shows preferential hypermyelination of small fibers in $Mpz^{T124M}$ mice compared to that in WT mice. (**G**) Ultrastructural analysis of periodicity shows that myelin sheaths are well compacted in $Mpz^{T124M}$ mice. Scale bars: 50 μm. (**H**) Myelin periodicity was slightly, but not significantly, increased in $Mpz^{T124M}$ mice. Nested one-way ANOVA [12 month old: F(2,8) = 1.702, p = 0.24; 18 month old: F(2,10) = 1.03, p = 0.392]. (**I**) Representative electron micrograph of an onion bulb. (**J**) Onion bulbs and supernumerary SC were rare in $Mpz^{T124M/T124}$ mice. They represented 0.2% of myelinated fibers in

6-month-old mice and did not progress over time. One-way ANOVA [2 month old: $F_{(2, 10)} = 2.285$, $p = 0.1523$; 6 month old: $F_{(2, 9)} = 5.779$, $p = 0.0243$; 12 month old: $F_{(2, 6)} = 3.628$, $p = 0.0927$; 18 month old: $F_{(2, 8)} = 2.620$, $p = 0.1333$]. (**K**) Representative electron micrograph of a myelin balloon formed by fluid inside the intraperiodic line of the myelin sheath. Note how the axon is compressed. **I** and **N** scale bars: 2 μm. $n$ (animals) ≥ 3 per genotype. *$p < 0.05$, **$p < 0.01$, by multiple-comparisons Tukey's *post hoc* tests after one-way ANOVA (**E** and **J**), after Nested one-way ANOVA. Graphs indicate means ± SEMs.

damage (**Figs 3C and S6**). Accordingly, degenerative features became apparent in $Mpz^{T124M}$ sciatic nerves starting at 2 months of age and progressively increased over time affecting at 18 months, 2% and 3.5% of myelinated fibers in $Mpz^{T124M/T124M}$ and $Mpz^{T124M/+}$ mice, respectively (**Fig 3D**). Beginning as early as 2 months of age in $Mpz^{T124M}$ mice, we observed the classic hallmarks of axonal degeneration, such as the detachment of compressed axons from the inner myelin loop leaving behind large periaxonal collars (**Fig 3E**). These are typical features of CMT1X neuropathy and could impair SC-to-axon communication [29,30]. We also observed axonal swelling in nerves from $Mpz^{T124M/T124M}$ mice. Some of the swollen axons contained dense axoplasma with an accumulation of organelles, mostly mitochondria and dark vacuoles, suggesting defects in axonal transport, whereas other swellings showed empty vacuoles (**Fig 3F and 3G**). In $Mpz^{T124M}$ mice 6 months of age and older, macrophages were detected that appeared to be clearing out degenerated fibers (**Fig 3J and 3J′**). We observed additional features of axonal degeneration, such as axonal regenerative clusters like those seen in samples from T124M-CMT2J patients and sciatic nerves from $Mpz^{T124M/T124M}$ and $Mpz^{T124M/+}$ mice at 12 and 18 months of age, respectively (arrows in **Figs S2C, S2D and 3I**). Regenerating axons were either amyelinated or hypomyelinated. Furthermore, in the axolemma from $Mpz^{T124M}$ nerves, we detected glycogenosomes (vacuoles containing glycogen) (**Fig 3H** arrowhead). Glycogenosomes are naturally present in axons of aged WT rats [31,32] but are more common in neuropathies associated with diabetes [33] and toxically induced neuropathy [34].

Remak bundles are formed by non myelinating SC ensheathing several small diameters axons (lower than 1μm). Ultrastructural image analysis of Remak bundles of $Mpz^{T124M}$ showed that, as in WT, the axons were homogenously wrapped by SC cytoplasmic extensions. We did not notice axonal abnormalities such as ploy-axonal pockets or degenerative axons in $Mpz^{T124M}$ [35] (**S7A and S7B Fig**). Quantitative analysis demonstrated that Remak bundles size (**S7C Fig**) and axon density (**S7D Fig**) were consistent among genotypes. Finally, we measured Remak bundles associated axonal diameter. In $MPZ^{T124M}$ mice, the mean axonal diameter and the proportion of large axons (1μm) were identical to WT, suggesting absence of axonal sorting defect [36]. Because Remak bundles associated axons are unmyelinated and SC wrapping those axons do not express P0 protein, despite a lightly active P0 promoter [37], these results suggest that axonal degeneration observed in $Mpz^{T124M}$ mice is dependent of P0 protein expression and of the presence of myelin sheath.

To further assess axonal degeneration, we crossbred $Mpz^{T124M}$ mice with two different transgenic reporter mouse lines. First, we crossed $Mpz^{T124M}$ mice with Thy1-YFP reporter mice, which express yellow fluorescent protein in a small proportion of neurons [38]. In $Mpz^{T124M/T124M}$ mice as young as 2 months old, we identified bulb-like structures (swellings) at the tips of degenerating axons. They occasionally had a fragmented appearance typical of that seen during Wallerian degeneration after nerve transection or crush (**Fig 3K**). At 10–11 months, $Mpz^{T124M/T124M}$ mice manifest dramatic axonal degeneration, and $Mpz^{T124M/+}$ mice also exhibit some axonal swelling and axonal degeneration (**Fig 3L**). Second, we crossed $Mpz^{T124M}$ mice with the ATF3-GFP reporter mouse line. Activating transcription factor 3 (ATF3) is a transcription factor rapidly expressed in neurons and SC after injury and axonal stress; intact adult neurons do not express ATF [39]. The expression of ATF3 after sciatic

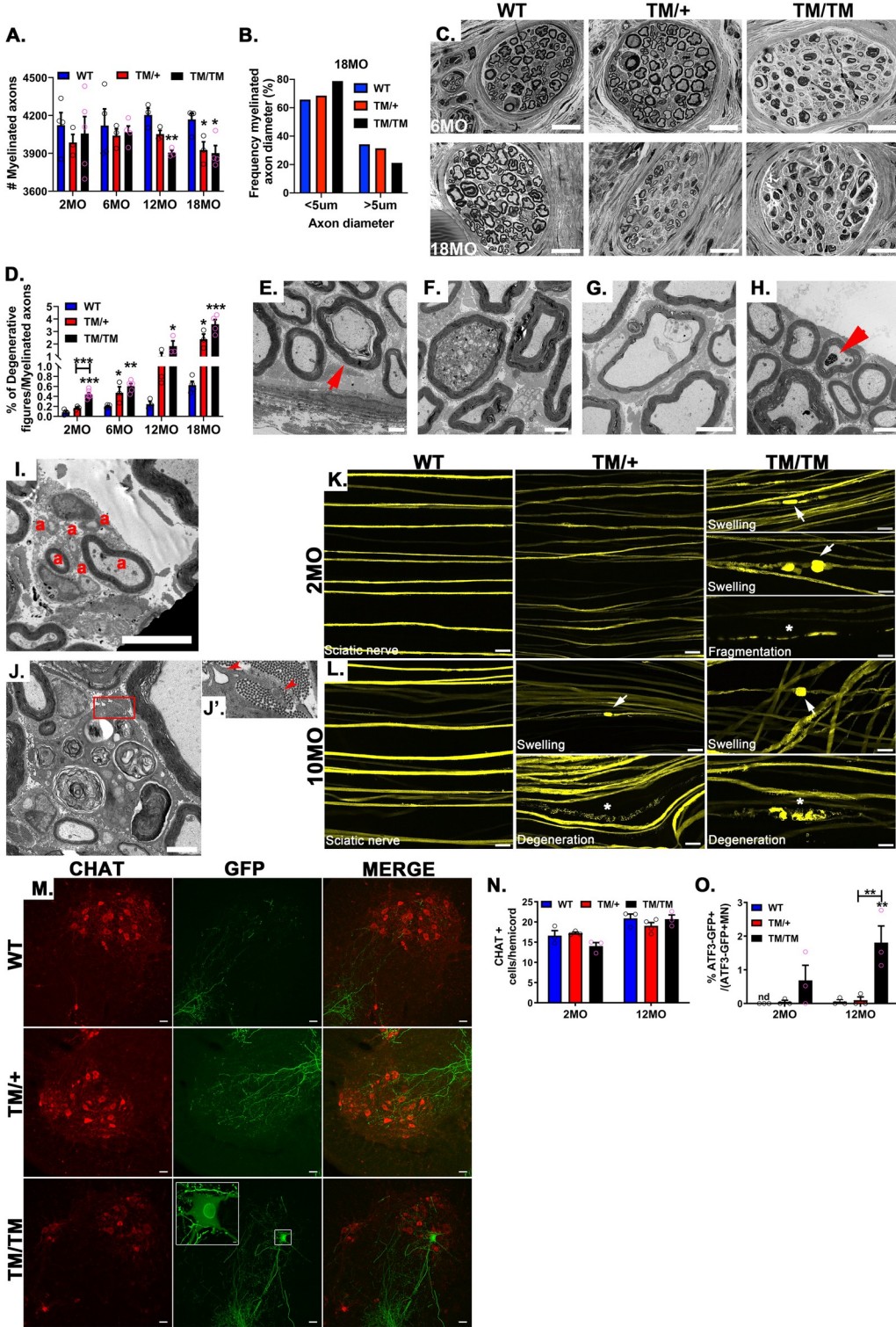

**Fig 3. P0T124M mutation causes axonal degeneration.** (**A**) Quantification of myelinated axons in whole sciatic nerves at 2, 6, 12, and 18 months of age. Starting at 12 months of age, the number of myelinated axons is significantly reduced in $Mpz^{T124M}$ mice. One-way ANOVA [2 month old: F (2, 9) = 0.2896, p = 0.7553; 6 month old: F (2, 8) = 0.1831, p = 0.8361; 12 month old: F (2, 6) = 15.97, p = 0.0040; 18 month old: F (2, 8) = 7.077, p = 0.0170]. (**B**) Frequency distribution of myelinated axons as a function of axonal diameter shows that big axons (>5 μm in diameter) are slightly more impacted than small axons (<5 μm), especially in 18-month-old $Mpz^{T124M/T124M}$ (TM/TM) mice. (**C**) Semithin cross sections of

digital (toe) nerves show presence of degenerative fibers in 6-month-old $Mpz^{T124M/T124M}$ mice. At 18 months of age, $Mpz^{T124M/+}$ and $Mpz^{T124M/T124M}$ mice exhibit degenerative fibers. Scale bars: 20 μm. (**D**) Quantification of degenerative figures in whole sciatic nerves at 2, 6, 12, and 18 months of age. One-way ANOVA: 2 month old [F (2, 10) = 37.30, p<0.0001; 6 month old: F (2, 9) = 11.47, p = 0.0033; 12 month old: F (2, 6) = 7.204, p = 0.0254; 18 month old: F (2, 8) = 25.66, p = 0.0003]. (**E to J**) Representative electron micrographs of degenerative axons observed in $Mpz^{T124M}$ mice. (**E**) Typical axonal cuffing with large periaxonal collar. (**F**) Swelled axon containing dark vesicles and mitochondria. (**G**) Empty axonal swelling. (**H**) Myelinated axon containing glycogenosomes (arrowhead). (**I**) Axonal regenerative clusters. Note the presence of hypomyelinated or amyelinated axons. "a" indicate axons (**J**) Degenerative fiber associated with a macrophage. (**J'**) Inset magnified 1.5× shows macrophage protrusions. **E**, **F**, **H**, and **J** scale bars: 2 μm; **G** and **I** scale bars: 5 μm. (**K and L**) Representative confocal microscopy images of WT and $Mpz^{T124M/T124M}$–Thy1-YFP sciatic nerves. (**K**) Note the presence of axonal swellings (arrows) and axonal degeneration (asterisks) in $Mpz^{T124M/T124M}$ at 2 months of age. (**L**) Axonal swellings (arrows) and axonal degeneration (asterisks) are observed in $Mpz^{T124M/+}$ and $Mpz^{T124M/T124M}$ mice at 10 months of age. Scale bars: 20 μm. (**M**) Representative confocal microscopy images of spinal cords sections (L3) from 12-month-old WT and $Mpz^{T124M/T124M}$–ATF3-GFP mice stained for choline acetyltransferase (CHAT; motoneurons) (red) and ATF3-GFP (green). Scale bars: 40 μm. High-magnification inset shows motoneuron expressing GFP under ATF3 promoter control. (**N**) Quantification of motoneurons (CHAT-positive cells) in hemicords of 2- and 12-month-old mice. One-way ANOVA [2 month old: F (2, 6) = 3.592, p = 0.0943; 12 month old: F (2, 7) = 1.188, p = 0.3596]. (**O**) Percentages of motoneurons expressing ATF3-GFP are increased in $Mpz^{T124M/T124M}$ mice at 2 and 12 months of age. One-way ANOVA [2 month old: F (2, 6) = 2.149, p = 0.1977; 12 month old: F (2, 7) = 14.42, p = 0.0033]. $n$ (animals) $\geq$ 3 per genotype. $^{*}p < 0.05$, $^{**}p < 0.01$ by multiple-comparisons Tukey's *post hoc* tests after one-way ANOVA. Graphs indicate means ± SEMs.

nerve injury prevents cell death and promotes neurite formation and elongation, leading to enhanced nerve regeneration through transcriptional induction of survival and growth-associated genes [40]. As expected, in 2- and 12-month-old $Mpz^{T124M/T124M}$–ATF3-GFP animals, ATF3-GFP was detected in motoneuron cell bodies (Choline acetyltransferase [ChAT] positive) from L1 to L5 spinal cord sections. However, in $Mpz^{T124M/+}$ mice, ATF3-GFP expression was marginal at 2 and 12 months of age (**Figs 3M, 3O, S8A and S8C**). Consistent with a peripheral neuropathy, neither the total number of motoneurons nor their morphology were significantly changed in $Mpz^{T124M}$ mice (**Figs 3N and S8C**).

Altogether, our findings indicate that the $Mpz^{T124M}$ mouse model closely replicates the clinical findings and axonopathy described in T124M-CMT2J patients. Even though P0 is expressed only by SC, we noticed only subtle changes in the $Mpz^{T124M}$ compact myelin sheath, suggesting primary axon damage and Wallerian-like degeneration.

## P0T124M mutation impedes N-glycosylation

A denaturing Western blot revealed that P0T124M has a lower relative molecular weight than the WT protein (P0WT) (23 kDa and 28 kDa, respectively) (**Figs S4, 4A and 4B**). The methionine residue in place of threonine 124, which is within the acceptor sequence for *N*-glycosylation, may impede P0 *N*-glycosylation, resulting in the observed shift in migration. To test this, we removed sugars of *N*-glycosylation via enzymatic digestion with endoglycosidase H (EndoH) or peptide N-glycosidase F (PNGaseF), which decreased the molecular weight of P0WT to 25 kDa. By contrast, P0T124M was insensitive to digestion by both enzymes, suggesting that P0T124M is not *N*-glycosylated. However, the lack of glycosylation is not sufficient to explain the observed shift in the molecular weight of P0T124M, as *N*-glycosylation accounts for only 3 kDa (**Fig 4A and 4B**). To confirm this experimentally, we transfected an HA-tagged version of P0N122S, a non-glycosylatable mutant into COS-7 cells, and observed that it migrated at a higher relative molecular weight (29 kDa, comparable to deglycosylated P0WT-HA) than HA-tagged P0T124M (27 kDa) (**Fig 4C**). Therefore, additional modifications, likely independent of *N*-glycosylation and specific to the T124M mutation, may be responsible for the altered migration of P0T124M.

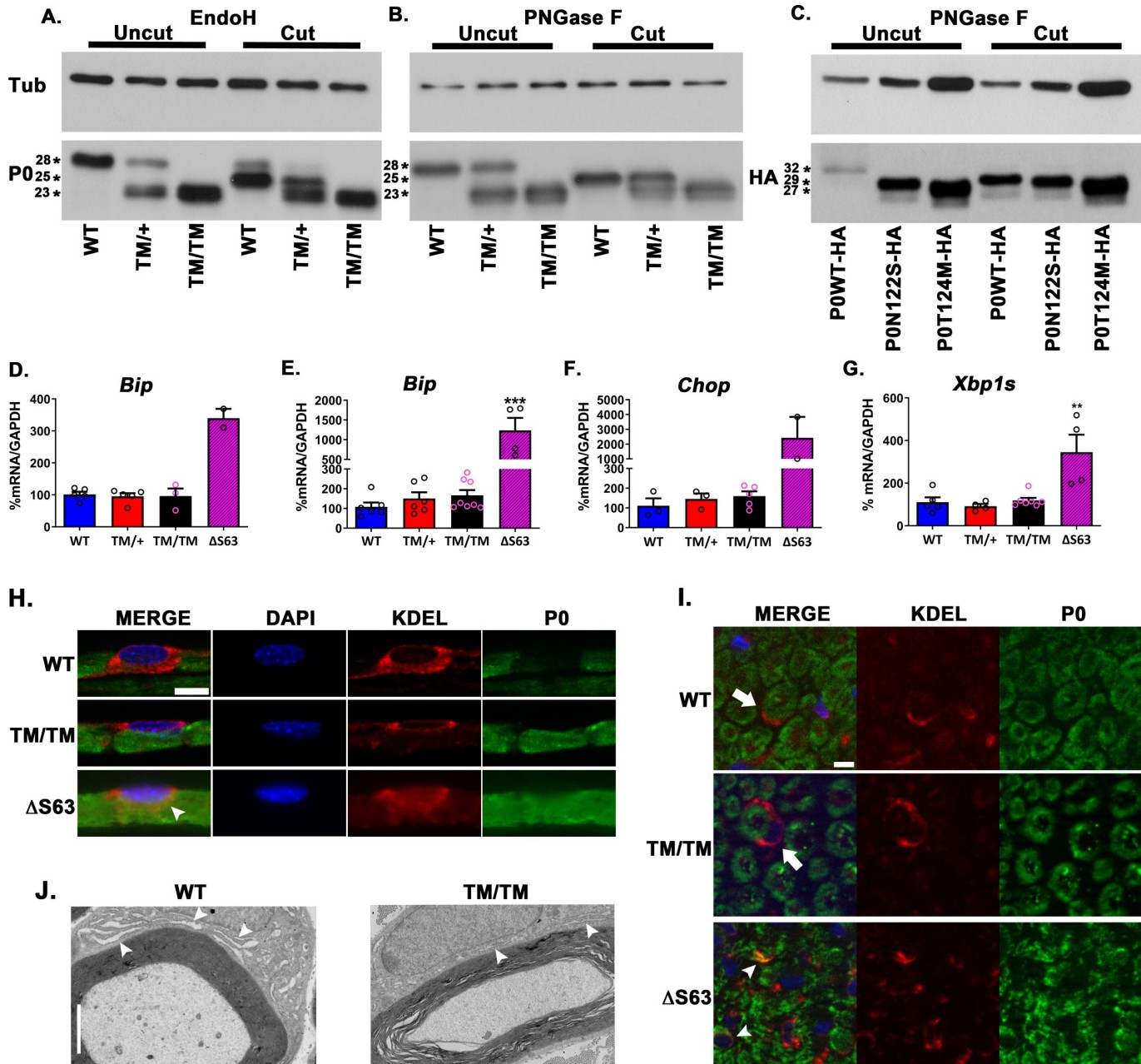

**Fig 4. T124M mutation is responsible for molecular P0 modifications but does not alter P0 trafficking or UPR activation.** Western blot analysis of P0 in sciatic nerve lysate from wild-type (WT), $Mpz^{T124M/+}$ (TM/+), and $Mpz^{T124M/T124M}$ (TM/TM) mice treated with (cut) or without (uncut) EndoH (**A**) and PNGase F (**B**). (**C**) COS-7 cells were transfected with P0-HA, P0T124M-HA, or P0N122S-HA. Samples were treated with (cut) or without (uncut) PNGase F and blotted with antibodies against HA. β-Tubulin (TUB) was used as a loading control. Asterisks indicate relative molecular weight. *Bip*, *Chop*, and *Xbp1* spliced mRNAs were measured at postnatal day 10 (**D**) and postnatal day 30 (**E, F, and G**) by RT-qPCR TaqMan assay. GAPDH mRNA was used for normalization. ΔS63 sciatic nerve mRNA was used as a positive control. One-way Anova [Bip P10: F (2, 10) = 0.07536, p = 0.9279; Bip P30: F (3, 20) = 20.72, p<0.0001; Chop P30: F (2, 8) = 0.7230, p = 0.5145; Xbp1s P30: F (3, 16) = 9.800, p = 0.0007]. Representative immunofluorescence images of sciatic nerve teased fibers (**H**) and cross sections (**I**) at postnatal day 30 and 2 months of age, respectively. Fibers and cross sections were stained for P0 (green), endoplasmic reticulum (ER) (KDEL, red), and DAPI (blue). Note that P0T124M, like P0WT, is not retained in the ER and reaches the myelin sheath (arrows). P0ΔS63 is retained in the ER (arrowheads) and does not reach the myelin sheath. (**J**) Representative electron micrographs of WT and $Mpz^{T124M/T124M}$ myelinating Schwann cell ER at 2 months of age. ER from $Mpz^{T124M/T124M}$ SC is not enlarged in comparison to ER from WT SC. Arrowheads indicate ER. Scale bars: 2 μm. *n* (animals) ≥ 3 per genotype. *$p < 0.05$, **$p < 0.01$, ***$p < 0.001$ by multiple-comparisons Tukey's *post hoc* tests after one-way ANOVA. Graphs indicate means ± SEMs.

## Pathogenesis of P0T124M does not involve the unfolded protein response

*N*-Glycosylation occurs in the endoplasmic reticulum and Golgi apparatus, and it is important for proper protein folding and for protein quality control [41]. The lack of *N*-glycosylation could impede P0T124M trafficking and generate endoplasmic reticulum stress. For example, the unfolded protein response is activated in two CMT1B mouse models: P0ΔS63 [26,42] and P0R98C [43]. However, the levels of mRNA for unfolded protein response markers (*binding immunoglobulin protein* [*Bip*], *C/EBP homologous protein* [*Chop*], *and X-box binding protein 1 spliced* [*Xbp1s*]) were not increased in sciatic nerves from *Mpz*^T124M mice (**Fig 4D–4G**). Moreover, the trafficking of P0T124M was not altered, as the mutant protein was not retained in the endoplasmic reticulum and was able to reach the plasma membrane (**Fig 4H and 4I**). Consistently, EM analysis did not reveal ER morphology alteration in *Mpz*^T124M SC (**Fig 4J**).

These results suggest that the unfolded protein response is not involved in the pathogenesis of T124M-CMT2J disease.

## P0T124M mutation perturbs axon-glia interactions

Paranodes are specialized regions for axon-glia interaction that permit the diffusion of metabolites, hormones, and water-soluble molecules from SC to the periaxonal space [44]. Defects in paranodes are associated with axonal degeneration [45–47], and P0 was shown to help maintain paranodal and nodal structures [48]. To detect potential defects in paranodal and nodal regions caused by the P0T124M mutation, they were stained with the markers Contactin associated protein 1 (CASPR) and pan-Neurofascin (NFASC) in isolated teased fibers. At 2 months of age, the paranodal region in *Mpz*^T124M fibers was longer than in WT fibers (**Fig 5A–5C**). The frequency distribution of CASPR clusters was shifted toward increased lengths in *Mpz*^T124M mice relative to WT mice. Nodal length trend to increase in *Mpz*^T124M/T124M mice (**Fig 5D**). Similar results were obtained from 12-month-old mice (**Fig 5E–5H**), confirming the elongation of paranodal and nodal areas in *Mpz*^T124M mice. Then, using electron microscopy on longitudinal ultrathin sections of WT and *Mpz*^T124M/T124M sciatic nerves at 12 month old, we examined ultrastructural morphology of nodes and paranodes. Consistently with CASPR staining, nodes were significantly wider in *Mpz*^T124M/T124M compared to WT (**Fig 5J**). We also noticed, in *Mpz*^T124M/T124M, some structural alteration of paranode such as messy, detached and inverted paranodal loops (**Fig 5I**). Such abnormalities were reported to be absent in WT nerves [48]. However, we occasionally noticed presence of disorganized paranodes also in WT, probably as consequence of poor fixation quality and potential osmotic stress.

P0 is also important for the creation of Schmidt-Lanterman incisures (SLI), another non-compact myelin domain crucial for the transit of signals and molecules between the outer and inner SC surface and the axon [49]. Morphological analyses of *Mpz*^T124M sciatic nerves revealed a significant increase in SLI in a time- and gene dose-dependent manner (**Figs 2B, 2C** asterisks, **6A–6D**, and **S2** asterisks). This was confirmed by staining for ACTIN, NFASC, and MAG, three proteins enriched in SLI, in isolated teased fibers from 2-, 6-, and 12-month-old animals (**Fig 6E, 6I, 6M and 6Q**). In mice as young as 2 months of age, the SLI in *Mpz*^T124M fibers were shorter (**Fig 6F, 6J, 6N and 6R**). Moreover, the morphology of SLI was also altered. Fibers from WT mice showed the expected series of single conical and clearly separated SLI, whereas the SLI in *Mpz*^T124M mice seem disorganized, more numerous, and sometimes fused together. By measuring the distance between two adjacent SLI, we confirmed that the number of SLI was increased in *Mpz*^T124M mice (**Fig 6G, 6K, 6O and 6S**). Finally, the number of SLI per 100 μm in *Mpz*^T124M/T124M mice was 2-fold higher than in WT mice (**Fig 6H, 6L, 6P and 6T**). These results suggest that P0T124M impairs the formation, structure, or function of SLI, which could ultimately result in lack of axonal metabolic support.

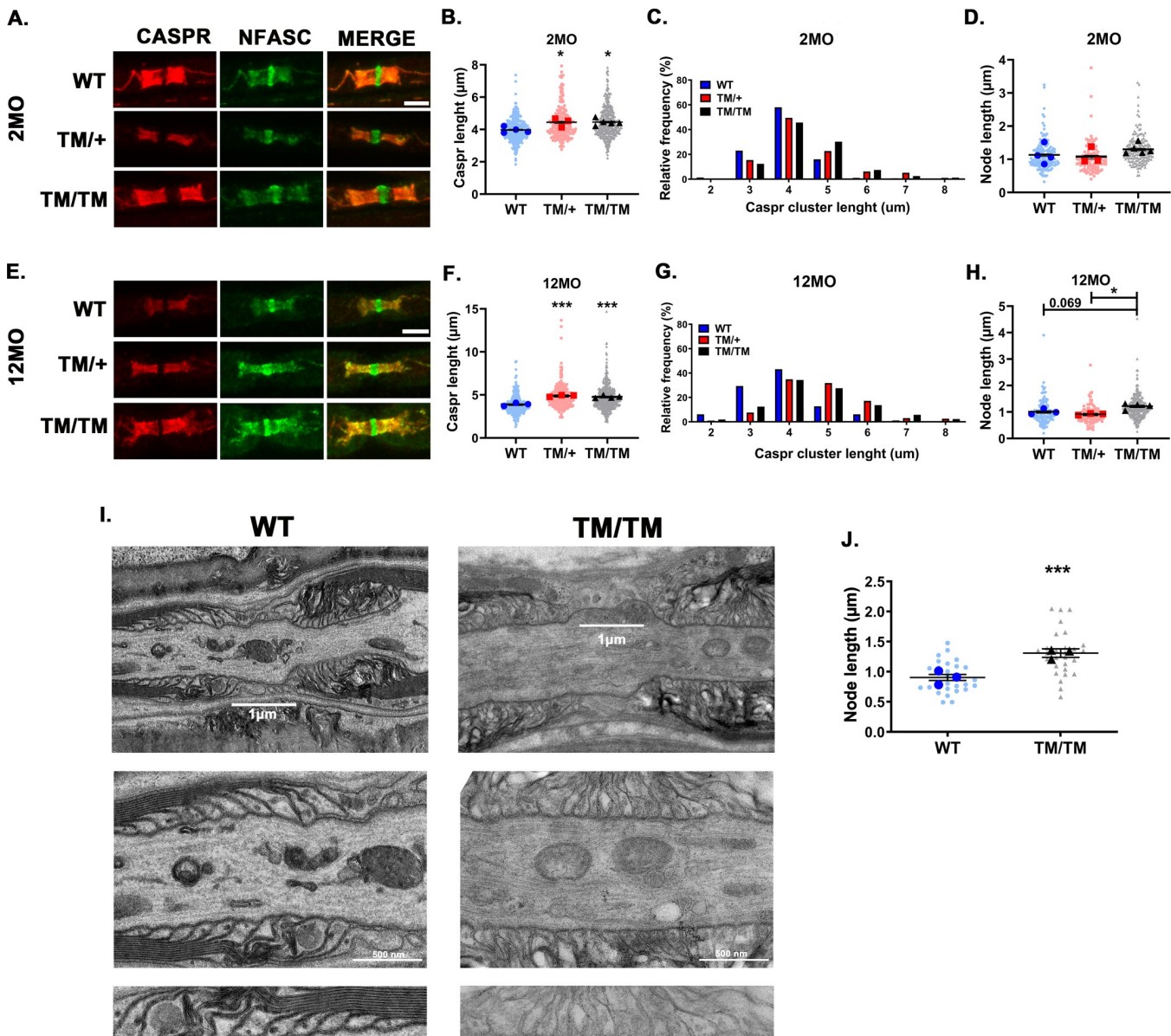

**Fig 5. P0T124M mutation alters nodes and paranodes.** Representative confocal pictures of sciatic nerve teased fibers stained with antibodies against the paranodal marker CASPR (red) and the paranodal and nodal marker pan-Neurofascin (NFASC; green) at 2 (**A**) and 12 (**E**) months of age. Scale bars: 5 μm. CASPR length quantifications at 2 (**B**) and 12 (**F**) months of age. Nested one-way ANOVA [2 month old: $F_{(2,9)}$ = 7.009, p = 0.0146; 12 month old: $F_{(2,7)}$ = 35.47, p = 0.0002]. Relative frequency distributions of paranodal (CASPR) length at 2 (**C**) and 12 (**G**) months of age. Nodal length quantifications at 2 (**D**) and 12 (**H**) months of age. Nested one-way ANOVA [2 month old: $F_{(2,9)}$ = 1.122, p = 0.3671, 12 month old: $F_{(2,7)}$ = 8.012, p = 0.0155]. At least 200 paranodes and 100 nodes per genotype were quantified at each time point. $n$ (animals) ≥ 3 per genotype. (**I**) Electron micrographs of ultrathin longitudinal WT and $Mpz^{T124M/T124M}$ sciatic nerve section at 12 months of age. In (**I**) electron micrographs represent nodes of WT and $Mpz^{T124M/T124M}$ sciatic nerves. (**J**) The quantification of nodal length shows significant widening of the node in $Mpz^{T124M/T124M}$. Magnifications of (**I**) show disorganized paranonal loops in $Mpz^{T124M/T124M}$ but well organized in WT. $n$ (animals) ≥ 3 per genotype, 7 to 11 nodes were counted per animals, for a total of 27 and 28 counted by genotype. Scale bare: top panel 1 μm, middle panel 500 nm, bottom panel 200 nm. \*\*\**p < 0.001* by multiple-comparisons Tukey's *post hoc* tests after Nested one-way ANOVA (**B, F, D, and H**) or Nested two-tailed Student's *t* test (**J**). Graphs indicate means ± SEMs.

Connexin 32 (CX32) is an important protein in the PNS. Similarly to P0T124M, CX32 mutations cause CMT1X, characterized by axonal abnormalities that precede demyelination [50]. CX32 is located in non-compact myelin domains (SLI and paranodes), where it forms

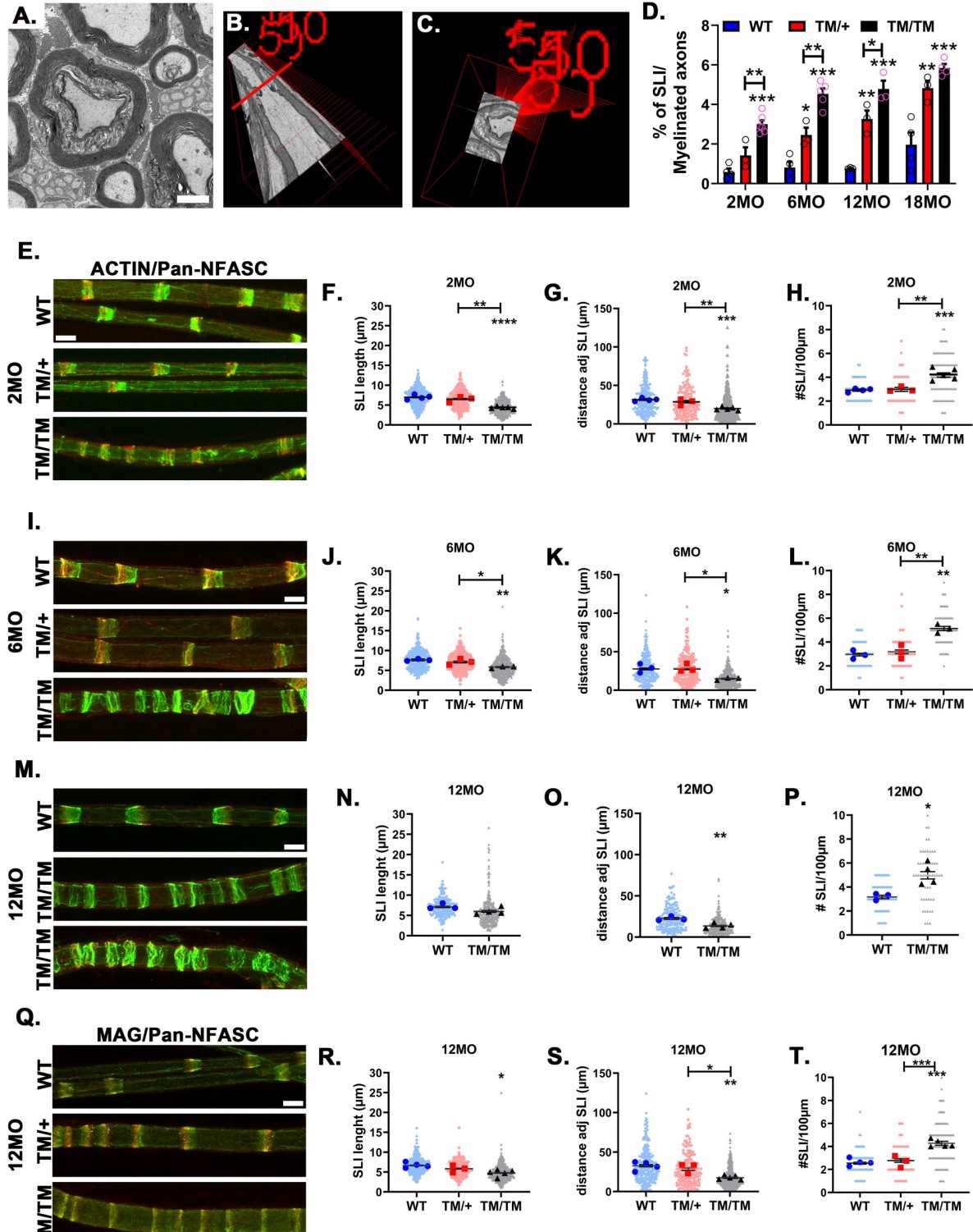

**Fig 6. P0T124M mutation alters SLI morphology, length, and distribution.** (**A**). Representative electron micrograph of Schmidt-Lanterman incisures (SLI) in sciatic nerve transverse section. SLI is illustrated by uncompacted myelin surrounded by two layers of compact myelin. (**B**) Three-dimensional electron micrograph showing myelinated axon with one SLI in longitudinal view. (**C**) Coronal view of the same SLI seen in B at the level of the red line. (**D**) Quantification shows an increased percentage of SLIs in nerves from mice harboring the P0T124M mutation ($Mpz^{\text{T124M}}$) at 2, 6, 12, and 18 months of age. One-way ANOVA [2 month old: F (2, 10) = 29.86, p<0.0001; 6 month old: F (2, 9) = 41.60, p<0.0001, 12 month old: F (2, 6) = 34.35, p = 0.0005; 18 month old: F (2, 8) = 21.71, p = 0.0006].

Representative confocal pictures of sciatic teased fibers stained with FITC-phalloidin (ACTIN, green) (**E, I, and M**) or anti-MAG antibodies (**Q**) (green) and anti-pan-NFASC antibodies (red) at 2 (**E**), 6 (**I**), and 12 (**M and Q**) months of age. SLI morphology is disrupted in $Mpz^{T124M}$ mice. Scale bar: 10 μm. Measurements of SLI length at 2 (**F**), 6 (**J**), and 12 (**N and R**) months of age. Nested one-way ANOVA [2 month old: $F_{(2,9)} = 32.16$, p <0.0001; 6 month old: $F_{(2,6)} = 11.18$, p = 0.0095 12 month old: $F_{(2,9)} = 6.447$, p = 0.0183]. Measurements of the distance between adjacent SLI at 2 (**G**), 6 (**K**), and 12 (**O and S**) months of age. Nested one-way ANOVA [2 month old: $F_{(2,9)} = 24.90$, p = 0.0002; 6 month old: $F_{(2,6)} = 9.382$, p = 0.0142; 12 month old: $F_{(2,9)} = 13.04$, p = 0.0022]. Quantifications of SLI number per 100 μm at 2 (**H**), 6 (**L**), and 12 (**P and T**) months of age. Nested one-way ANOVA [2 month old: $F_{(2,9)} = 20.29$, p = 0.0005; 6 month old: $F_{(2,6)} = 22.81$, p = 0.016; 12 month old: $F_{(2,9)} = 28.25$, p = 0.0001]. At least 207 SLI per genotype were quantified at each time point. $n$ (animals) ≥ 3 per genotype. $^*p < 0.05$, $^{**}p < 0.01$, $^{***}p < 0.001$ by multiple-comparisons Tukey's *post hoc* tests after Nested one-way ANOVA (**D, F, G, H, J, K, L, R, S, and T**) or by two-tailed Student's *t* test (**N, O, and P**). Graphs indicate means ± SEMs.

gap junctions between the layers of the SC myelin sheath, providing a fast diffusive radial path between the abaxonal and adaxonal areas [51]. Although we were not able to reproducibly detect CX32 in SLI, we quantified the fluorescence intensity of CX32 staining in paranodes of WT and $Mpz^{T124M}$ mice at 2 months of age (**Fig 7A**). Intensity of CX32 staining was similar in $Mpz^{T124M/+}$ and WT but tended toward decreased in nerves from $Mpz^{T124M/T124M}$ mice (**Fig 7B and 7E**). CX32 was also mislocalized in some $Mpz^{T124M}$ fibers. While 90% of paranodal CX32 colocalized with CASPR in WT fibers, 20% of $Mpz^{T124M/+}$ fibers and 30% of $Mpz^{T124M/T124M}$ fibers showed CX32 outside the paranodal area (**Fig 7C**). To confirm CX32 mislocation in $Mpz^{T124M}$ mutants, we costained teased fibers for CX32 and potassium voltage-gated channel 1.1 (Kv1.1), a marker of the juxtaparanodal region (**Fig 7D**). CX32 was expressed in juxtaparanodes (i.e., colocalized with Kv1.1) in only 15% of WT fibers but in 20% and 35% of fibers from $Mpz^{T124M/+}$ and $Mpz^{T124M/T124M}$ mice, respectively (**Fig 7F**). It is tempting to speculate that the altered CX32 distribution contributes to the axonopathy observed in $Mpz^{T124M}$ mice.

Altogether, our results indicate a perturbation of the areas of non-compact myelin where exchange and communication between SC and axons are thought to occur. These alterations could lead to deficient communication and support from SC to axons.

## ATP and NAD$^+$ but not glycolysis are decreased in $Mpz^{T124M}$ mice

Because of the alterations at axon-glia exchange areas, we reasoned that metabolite transport from SC to axons would be disturbed in $Mpz^{T124M}$ mice. We studied the steady state levels of key metabolites for SC and axon functions by mass spectrometry in sciatic nerves from 12-month-old mice (**Fig 8A–8H**). ATP, the main energy source for neurons, decreased by 30% in $Mpz^{T124M/T124M}$ mice (**Fig 8A**). In CNS, glycolytic activity in myelinating glia is fundamental for axonal energetics and survival [52–57]. We wondered if defect in glycolysis could explain ATP decrease in $Mpz^{T124M}$ PNS. However, we did not notice significant differences between WT and $Mpz^{T124M/T124M}$ mice in the amounts of glucose-6-phosphate (G6P) (**Fig 8B**), lactate (**Fig 8C**), or pyruvate (**Fig 8D**). Interestingly, there was striking and significant reduction (-55%) in the amount of oxidized nicotinamide adenine dinucleotide (NAD$^+$) in sciatic nerves from $Mpz^{T124M/T124M}$ mice (**Fig 8E**) along with a trend toward less reduced NAD (NADH) (**Fig 8F**). NAD$^+$ and NADH are involved in many cellular and biological functions such as energy metabolism, mitochondrial function, and redox state [58], each of which is important for axonal physiology. Moreover, NAD$^+$ balance is crucial for axonal survival. Cleavage of NAD$^+$ by SARM1 (sterile alpha and toll/interleukin 1 receptor motif-containing 1) into nicotinamide (NAM) and adenosine phosphoribose (ADPR) leads irremediably to axonal degeneration [59–61]. Although ADPR expression appears to be normal in $Mpz^{T124M/}$ nerves (**Fig 8G**), it is rapidly converted to cyclic ADPR (cADPR), making it difficult to obtain a reliable measurement. However, NAM expression shows a trend toward increased levels in $Mpz^{T124M/T124M}$ mice (**Fig 8H**). This combined with the reduced levels of NAD$^+$ suggest an involvement of SARM1 in the axonal degeneration observed in $Mpz^{T124M}$ mice.

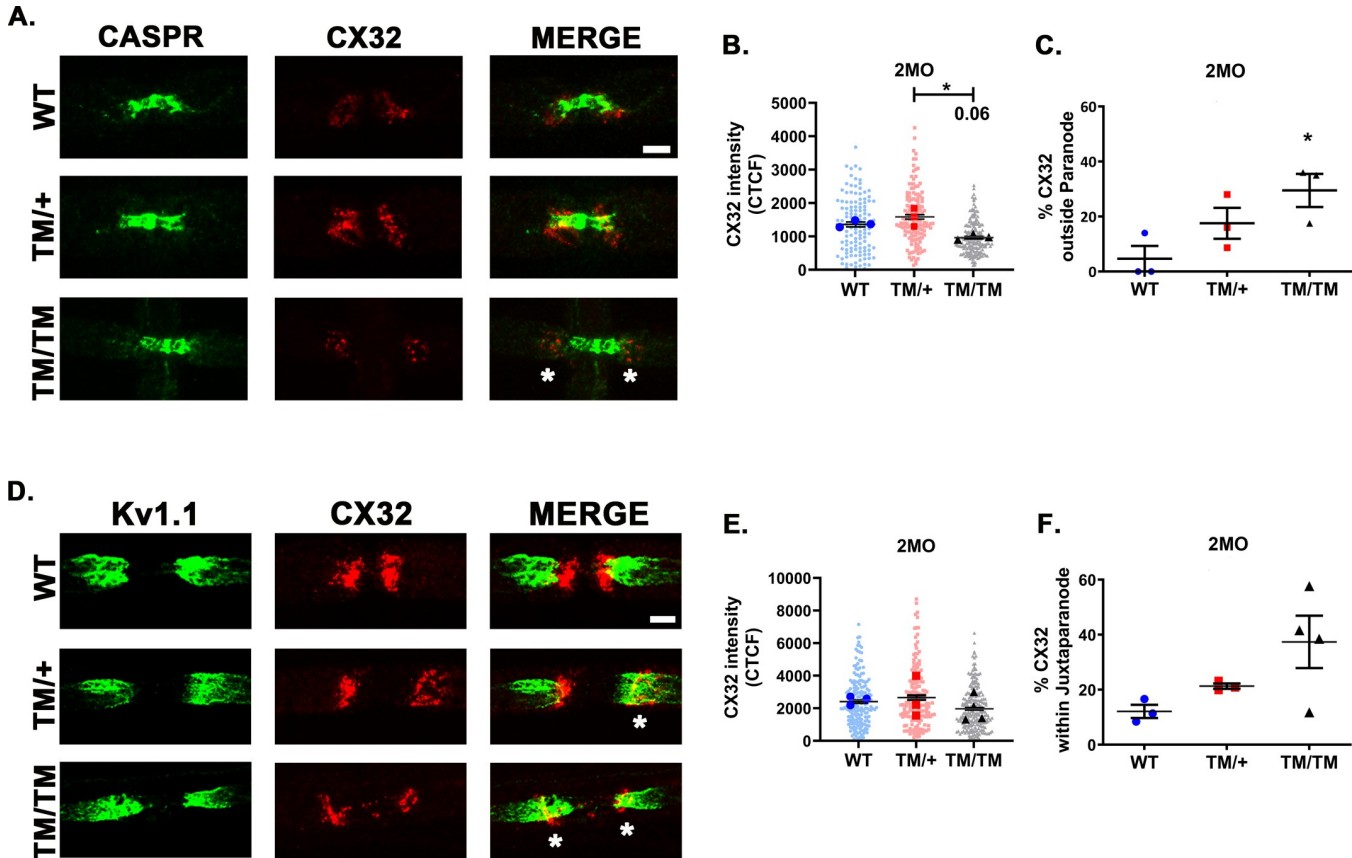

**Fig 7. CX32 expression is altered in *Mpz*^T124M nerves.** Representative confocal pictures of sciatic nerve teased fibers stained with antibodies against connexin 32 (CX32) (red) and CASPR (green) (**A**) or Kv1.1 (green) (**D**) from mice at 2 months of age. Asterisks indicate CX32 signal located at juxtaparanode. Scale bars: 5 μm. (**B and E**) Quantifications of the intensity of CX32 staining (measured by corrected total cell fluorescence methodology). Nested one-way ANOVA [(**B**) $F_{(2,6)}$ = 9.193, p = 0.0149; (**E**) $F_{(2,7)}$ = 0.559, p = 0.594]. Quantifications of CX32 signals expressed at juxtaparanodes: not colocalized with CASPR (**C**) and colocalized with Kv1.1 (**F**). One-way ANOVA [(**C**) $F_{(2,6)}$ = 0.0495, p =; (**E**) $F_{(2,7)}$ = 3.579, p = 0.0850]. CX32 intensity was quantified from at least 122 para- and juxtaparanodes per genotype. *n* (animals) ≥ 3 per genotype. *p < 0.05 by multiple-comparisons Tukey's *post hoc* tests after Nested one-way ANOVA (**B and E**) or one-way ANOVA (**C and F**). Graphs indicate means ± SEMs.

## Axonal transport and mitochondrial disruption in *Mpz*^T124M mice

Because ATP levels were decreased in *Mpz*^T124M mice and because neurons are dependent on mitochondria for ATP homeostasis and long-term integrity, we posited that the *Mpz*^T124M mice would exhibit axonal mitochondrial defects. We stained mitochondria in sciatic nerve teased fibers with anti-HSP60 (heat shock protein 60) antibodies (**Fig 8I and 8J**). By co-staining with neurofilaments and using confocal microscopy, we focused our attention to axonal mitochondria. In *Mpz*^T124M fibers, axonal mitochondria appeared fragmented. We observed a global decrease of axonal mitochondrial surface area in *Mpz*^T124M mutants compared to that in WT (**Fig 8K**). We also noticed, starting at 2 months of age, the presence of clustered axonal mitochondria in approximately 4 to 5% of *Mpz*^T124M axons, suggesting a defect in fast axonal transport (**Fig 8L**). Moreover, at 12 months of age, large axonal mitochondria were present in 1 and 10% of *Mpz*^T124M/+ and *Mpz*^T124M/T124M fibers respectively (**Fig 8J, arrows and 8M**). Large mitochondria could reflect clusters of small mitochondria or degenerative swelling. Altogether, our results indicate a defect in the transport and, potentially, functionality of axonal mitochondria in sciatic nerves in *Mpz*^T124M mice that in turn could impair ATP production, thereby leading to axonal death.

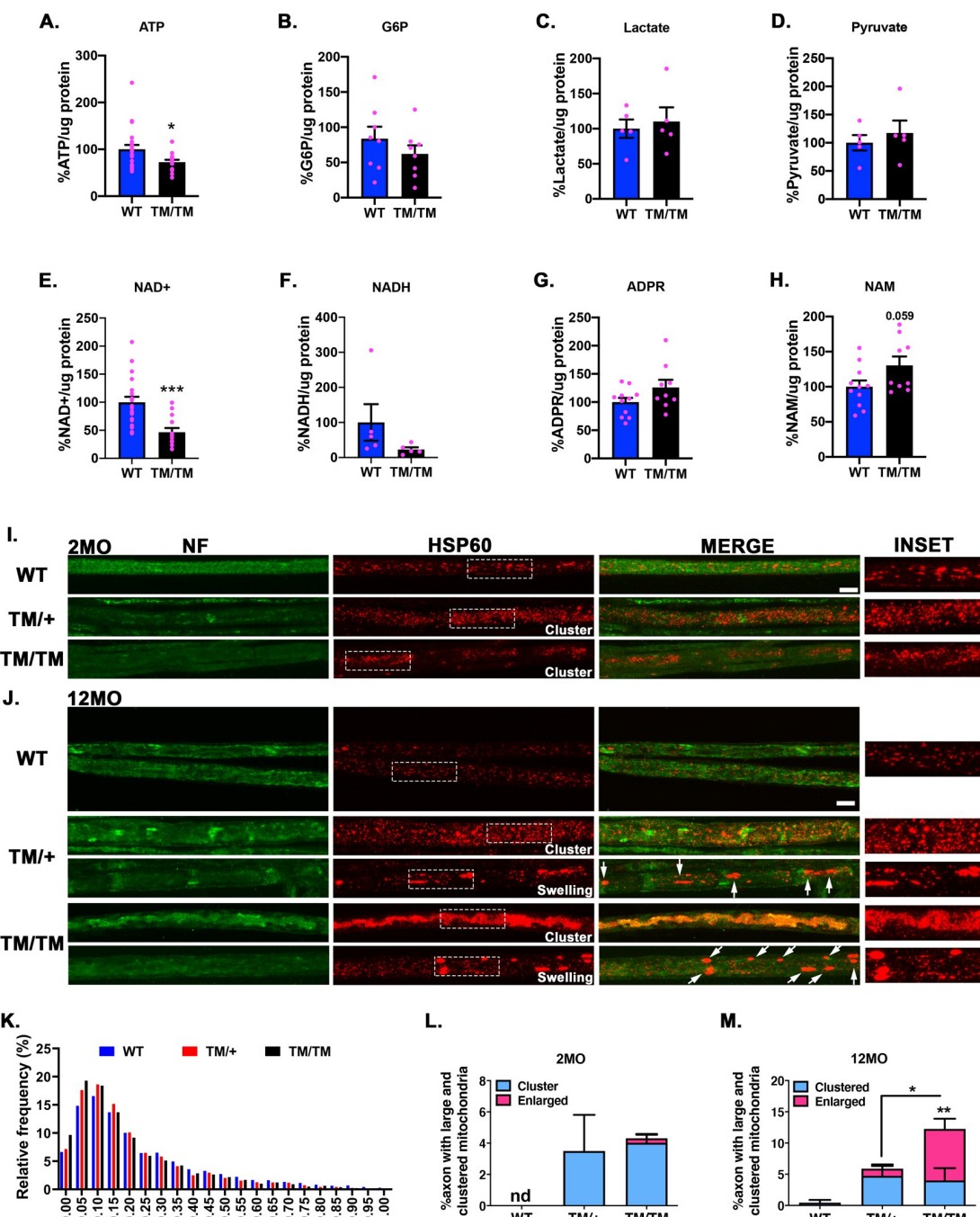

**Fig 8. Metabolic and axonal mitochondrial impairments in $Mpz^{T124M}$ mice. (A to H)** Measurements of metabolites involved in axonal energy and degeneration programs. ATP (**A**), G6P (**B**), lactate (**C**), pyruvate (**D**), NAD$^+$ (**E**), NADH (**F**), ADPR (**G**), and NAM (**H**) were quantified via liquid chromatography-tandem mass spectrometry of sciatic nerves from 12-month-old mice. *n* (animals) ≥ 5 per genotype. (**I and J**) Representative confocal pictures of sciatic nerve teased fibers stained with antibodies against mitochondrial protein HSP60 (red) and axonal marker neurofilament (NF) (green) from mice at 2 (**I**) and 12 (**J**) months of age. Arrows indicate swelled mitochondria. Scale bars: 10 μm. High magnification insets show mitochondria. (**K**) Relative frequency distributions of mitochondrial surface area (μm²) at 2 months of age. Nodal length Surface area was quantified from at least 1,096 mitochondria per animal. Proportions of axons with clustered and enlarged mitochondria at 2 (**L**) and 12 (**M**) months of age. One-way ANOVA [clustered mitochondria at 2 month old: F (2,6) = 2.558, p = 0.1573 and at 12MO month old: F (2,7) = 1.812, p = 0.2322; enlarged mitochondria at 12 month old F (2,7) = 14.43, p = 0.0033]. At least 263 axons per genotype were imaged. *n* (animals) ≥ 3 per genotype. *p < 0.05, **p < 0.01, ***p < 0.001 by two-tailed Student's *t* test (**A** to **H**) and by multiple-comparisons Tukey's *post hoc* tests after one-way ANOVA (**K** to **M**). Graphs indicate means ± SEMs.

## Discussion

We generated a genetically authentic knock-in mouse model of the human inherited neuropathy CMT2J caused by the T124M mutation encoded in the *MPZ* gene. In-depth characterization of these mice demonstrated that mutants closely replicate the axonopathy and other clinical aspects of T124M-CMT2J patients: an adult-onset progressive motor and auditory neuropathy with a marked reduction of CMAP, but only a slight decrease of NCV at later stages; reduced numbers of myelinated axons and the presence of regenerative clusters, but also well compacted myelin sheaths with near normal packing and periodicity with only occasional onion bulbs and supernumerary SC [14,62,63]. Although the presence of onion bulbs may suggest sporadic demyelination in *Mpz*$^{T124M/T124M}$ mice, our findings, together with the clinical features of the disease and the fact that changes in axonal changes preceded those in compact myelin, suggest that demyelination is not likely to trigger axonal degeneration in these mice.

Despite the similarities, we also identified differences between *Mpz*$^{T124M}$ mice and patients. The mouse axonal neuropathy appears to be milder than its human counterpart. Humans present with an axonal loss of 50–70% [14,15], whereas the loss is around 10% in mice. The allelic pattern is also different between mice and humans. T124M-CMT2J is an autosomal dominant disease. All CMT2J patients referenced were heterozygous for the P0T124M mutation except for one patient that was homozygous for the mutation; this patient was severely affected and died at the age of 44 years [62,63]. Although *Mpz*$^{T124M/+}$ mice similarly develop the disease, the human disease phenotype was more faithfully recapitulated in *Mpz*$^{T124M/T124M}$ mice. The milder severity and allelic pattern discrepancy are common observations in CMT2 knock-in mouse models [64] and could be explained by several factors, including the shorter life span and shorter axonal length in mice.

Another difference between humans and mice is myelin sheath thickness. In patients, the thickness of sural nerve myelin is varied. In general, CMT2J patients myelin sheath are thinner than healthy control, but some patients have normal or even thickened myelin sheaths [14,15]. In *Mpz*$^{T124M}$ mice, we did not observe hypomyelination. Myelin sheath thickness was normal until 6 months of age, and at 12 months of age, *Mpz*$^{T124M}$ fibers were slightly hypermyelinated.

Homophilic interactions of P0 proteins are fundamental for myelin compaction. P0 interacts with itself via the protein backbone, and P0 carbohydrates may contribute to homophilic adhesion [65,66]. Even though the P0T124M mutation abolished P0 *N*-glycosylation, the myelin sheath was relatively well compacted. Moreover, *in vitro* experiments have shown that the T124M mutation does not interfere with P0 adhesiveness [67]. Interestingly, the P0T124A and P0N122S mutations, which similarly abolish native *N*-glycosylation, are responsible for late-onset CMT1B with only moderate reduction of NCV [7,68,69], and like P0T124M, these mutated P0 proteins retain their adhesive propriety *in vitro* [70]. This suggests that the P0 protein backbone is generally sufficient for P0 homophilic adhesion and for myelin compaction. On the other hand, we cannot exclude the possibility that the absence of *N*-glycan has subtle consequences for P0 adhesion and myelin stability, which may explain the observed onion bulbs and hypermyelination. Future analyses of *Mpz*$^{T124M}$ myelin periodicity by X-ray diffraction would answer this question.

Notably, the P0T124M mutation altered non-compact myelin, which comprises domains essential for SC-to-axon interactions. Indeed, in *Mpz*$^{T124M}$ fibers, SLI were shorter, more numerous, and had a disorganized structure. The P0T124M mutation also altered paranodal structures. The aberrant CASPR immunoreactivity we observed could disrupt paranodal–axo-glial interactions [45]. Further experiments will be necessary to examined in detail septate like junction in *Mpz*$^{T124M}$. Moreover, the dialogue between SC and axons is likely impaired in

fibers with large periaxonal collars, as the adaxonal SC membrane cannot contact the underlying axons. Altogether, our results strongly suggest that axonal degeneration in $Mpz^{T124M}$ mice is caused by the disruption of the physical and molecular communication between glia and axons.

How P0 is involved in neuroglial interactions and signaling at SLI and paranodes is an important but unanswered question. P0 is not a structural component of SLI, but it appears to be necessary for SLI formation via an unknown mechanism [49,71]. Moreover P0, via interaction with NFASC, was shown to be directly involved in paranode maintenance [48]. Future experiments are needed to determine if the P0T124M mutation disrupts this interaction. It is tempting to speculate that the lack of P0 *N*-glycans as a result of the T124M mutation is detrimental for SC-to-axon interactions. Indeed, *N*-glycans modifications (GlcNAc-6-*O*-sulfation) are directly involved in the maintenance of paranodal organization in the PNS [47].

Because of the defect in SC-to-axon exchange areas and because myelinating glia and axons are metabolically linked [55,72,73], we suspected that the metabolic support from SC to axons is impaired in $Mpz^{T124M}$ mice. This hypothesis is further reinforced by the reduced ATP levels observed in nerves from $Mpz^{T124M}$ mice. In the CNS, oligodendrocyte glycolysis through lactate production is indispensable for producing the ATP needed for axonal activity and survival [52,53,56,57]. However, despite the reduced ATP and NAD$^+$ levels, which are crucial for glycolysis and oxidative phosphorylation [59,74], glucose uptake (G6P) and glycolysis (lactate and pyruvate) did not appear to be altered in $Mpz^{T124M}$ nerves. In the PNS, SC shift towards glycolysis in favor of lactate production to sustain axons after acute injury [75]. However, the lactate and glycolysis pathways play a limited role in axon survival in the PNS under physiological conditions [54,75–77]. Other metabolic pathways, such as lipid metabolism, may be more important for axonal health in the PNS [54]. Thus, $Mpz^{T124M}$ mice described here represent a highly valuable model to further our understanding of SC-to-axon metabolic interactions, and the identification of altered metabolic pathways could pinpoint potential therapeutic targets [78–81].

The mitochondrial defects observed further support the notion that P0T124M disrupts the metabolic support of axons. Neurons depend on mitochondrial ATP production to survive and are, therefore, particularly vulnerable to changes in mitochondrial morphology and connectivity. Our results suggest that the perturbation of SC-to-axon support that results from the P0T124M mutation has deleterious consequences for the morphology of axonal mitochondria, illustrated by a global reduction of axonal mitochondrial surface (fragmentation) and by the presence of clustered and swollen mitochondria. The relationship between mitochondrial fragmentation and bioenergetics is bidirectional because mitochondrial fragmentation decreases ATP production, which in turn induces mitochondrial fragmentation [82]. The presence of clustered and swollen mitochondria in the nerves from $Mpz^{T124M}$ mice also hints to a potential impairment in fast axonal transport, to which axons, because of their considerable length, are particularly sensitive [83]. A disruption of energy-dependent axonal transport is compatible with the distal dying back mechanism of axonal degeneration observed in $Mpz^{T124M}$ mice. Furthermore, axonal transport defects and clustered and swollen mitochondria are hallmarks of degenerating axons in myelin proteolipid lipid (PLP)-null mice [84] and P0-CNS mice, a transgenic mouse in which PLP expression was artificially substituted by P0 in oligodendrocyte myelin [85].

The observed reduction of NAD+ level in $Mpz^{T124M}$ sciatic nerves is more likely to reflect axonal NAD+ status than SC one. Indeed, decrease of NAD+ in SC is associated with hypomyelination and SC dedifferentiation without axonal damage [86]. Myelination and SC differentiation are not significantly altered in $Mpz^{T124M}$ nerves. To the contrary, reduction of axonal NAD+ level is demonstrated to trigger axonal degeneration. Thus mitochondrial dysfunction

[61,87] and decreased levels of NAD$^+$ coupled to the trend towards increase in NAM levels [60] point to a possible role for SARM1 in the axonal degeneration observed in $MPZ^{T124M}$ mice. SARM1 is considered a very interesting therapeutic target to limit axonal degeneration in many neurodegenerative diseases, and such approaches may also apply to T124M-CMT2J.

## Conclusion

The authentic mouse model for T124M-CMT2J we generated shows that the P0T124M mutation alters paranode, SLI, and SC gap junction structures, possibly resulting in a defect of trophic support from SC to axons and energetic failure. We suspect that the initial ATP deficiency results in axonal mitochondrial dysfunction, which in turn worsens the axonopathy. Axonal transport is slowed, leading to a vicious cycle that ultimately results in axonal degeneration with potential involvement of the SARM1 cascade (**S9 Fig**). The fidelity of the $Mpz^{T124M}$ mouse model to CMT2J disease observed in patients and the understanding of the pathomechanisms underlying SC–axon communication defects will enable us to explore new therapeutic strategies for T124M-CMT2J disease as well as other CMT and neurological diseases that involve axonal degeneration.

## Materials and methods

All experiments performed on mice were conducted in accordance with experimental protocols approved by the Institutional Animal Care and Use Committees of Roswell Park, University at Buffalo, and San Raffaele Scientific Institute.

### Animals

The P0T124M mutation was targeted to a single *Mpz* allele by homologous recombination as described by Saporta *et al.* [43] for the generation of P0R98C mice. The mutation was introduced into *Mpz* exon 3 of a 129S2 genomic clone by site-directed mutagenesis and confirmed by sequence analysis. Fragments of this clone were ligated into a construct containing the neomycin resistance gene (*neoR*) flanked by loxP sites (**S1A Fig**). The T124MneoLP construct was electroporated into TBV2 (129S2 strain) embryonic stem cells as described previously by Nodari *et al.* [88]. After confirming homologous recombination by Southern blot analysis with probes recognizing the sequence flanking either the long or the short arm of the construct, positive embryonic stem cell clones were injected into blastocysts of host wild-type animals to obtain chimeras. One chimera transgene was germline transmitted, and the mouse harboring the transgene was crossed with CMV-Cre mice (JAX#006054) to excise the *neoR* cassette (**S1A Fig**). To confirm that the mutation was present as expected, RNA from the sciatic nerves of $Mpz^{T124M}$ mice was isolated and reverse transcribed. The full coding sequence of *Mpz* cDNA was amplified by PCR (forward (F), 5′-ATGGCTCCCGGGGCTCCC-3′; reverse (R), 5′-CTATTTTCTTATCCTTGCGAG-3′), cloned, and sequenced (**S1 Data**). The $Mpz^{T124M}$ line was established in two different backgrounds: C57BL/6N and FVB/N. All results were obtained using the $Mpz^{T124M}$ C57BL/6N line, except for those from auditory system (**Fig 1L, 1J and 1K**) and morphology (**S4 Fig**) experiments, in which the $Mpz^{T124M}$ FVB/N line was used. Progeny used in this study were N3–N10 congenics in either the C57BL/6N or FVB/N background. For the detailed characterization of axonal morphology and neuronal stress, $Mpz^{T124M}$ mice were crossed with B6.Cg-Tg(Thy1-YFP)HJrs/J mice (JAX# 003709), here called Thy1-YFP, and ATF3-GFP mice. ATF3-GFP mice were kindly provided by Dr. Clifford J. Woolf (Harvard Medical School).

Animals were housed no more than five animals per cage with a 12-h light/dark cycle. Mutant and control littermates from either sex were sacrificed at the ages indicated in the text.

## Genotyping and PCR primers

Genotyping was performed by PCR analysis of genomic DNA extracted from toe clips or ear punches. The P0T124M mutation introduced an extra NSP1 restriction enzyme digestion site in *Mpz* exon 3 (**S1B and S1C Fig**). PCR primers MpzEx3F (5′-CGATGAGGTGGGGGGCCTT-CAA-3′) and MpzEx3R (5′-ATAGAGCGTGACCTGAGAGG-3′) generate a 169 base pairs (bp) amplimer. After NSP1 digestion, the wild-type allele migrates at 114 bp + 55 bp, whereas the T124M allele migrates at 106 bp + 8 bp + 55 bp. The 8-bp band was not detected on the PCR gel (**S1D and S1E Fig**). In addition to the NSP1 strategy, a loxP site was also used to geno-type *Mpz*$^{T124M}$ mutants. PCR primers MpzInt3 Fw (5′-TCAAAGAGGGTGTCAGGGAG-3′) and MpzInt3 Rv (5′-GTGGCCCAGATTGGTCTTTA-3′) generate a 355- or 305-bp amplimer for loxP or wild type, respectively. For ABR and cochleogram experiments, cadherin 23 (*cdh23*) polymorphisms were genotyped. PCR primers Cdh23 Fw (5′-ATCATCACGGA-CATGCAAGA-3′) and Cdh23 Rv (5′-AGCTACCAGGAACAGCTTGG-3′) generate a 315-bp amplimer. After digestion with BSRI, the C57BL/6N allele migrates at 238 bp + 77 bp, whereas the FVB/N allele is not sensitive to BSRI digestion. Instead, after BSAWI digestion, the FVB/N allele migrates at 236 bp + 79 bp, whereas the C57BL/6N allele is not sensitive to BSAWI diges-tion. Only animals with at least one *cdh23* FVB/N allele were used for experiments. Thy1-YFP mouse genotyping was performed as described by Jackson Laboratory (https://www.jax.org/strain/003782). PCR primers Thy-YFP Fw (5′-ACAGACACACACCCAGGACA-3′) and Thy-YFP Rv (5′-CGGTGGTGCAGATGAACTT-3′) generate a 400-bp amplimer. PCR primers for internal control Fw (5′-CTAGGCCACAGAATTGAAAGATCT-3′) and internal control Rv (5′-GTAGGTGGAAATTCTAGCATCATCC-3′) generate a 300-bp amplimer. PCR primers ATF3-GFP Fw (5′-CAACATCCTGTTCGGCAACCAA-3′) and ATF3-GFP Rv (5′-TCCACGCGGTACACGAACTT-3′) generate a 244-bp amplimer.

## Isolation of genomic DNA for genotyping

Toes and ear punches were digested in 75 μL of 25 mM NaOH-0.2 mM EDTA at 95°C for 45 min and neutralized with 75 μL of 40 mM Tris-HCl (pH 5.5). Samples were centrifuged and used for PCR reactions.

## Cell culture and transfection

COS-7 cells were obtained from the American Type Tissue Collection. Cells were grown in low-glucose Dulbecco's modified Eagle's medium supplemented with 10% fetal bovine serum and antibiotics (100 μg/ml penicillin-streptomycin) in a humidified atmosphere containing 5% $CO_2$ at 37°C. For transient transfection, Lipofectamine 2000 (Invitrogen) and 4 μg of plas-mid DNA were incubated separately in Opti-MEM (Gibco-BRL) for 5 min at room tempera-ture and then combined for another 20 min. When cells reached approximately 80% confluence, they were washed with Opti-MEM and then incubated with the combined Lipo-fectamine-DNA solution for 6 h at 37°C. The cells were then washed once with HBSS (free of calcium and magnesium) and incubated for 3 days in medium at 37°C before being processed for Western blotting.

## DNA constructs

HA-tagged P0WT and P0N122S plasmids used in these experiments were previously gener-ated by Dr. Elisa Tinelli as described by Penutto *et al*(42). HA-tagged P0T124M was generated by site-directed mutagenesis using a QuikChange II site-directed mutagenesis kit (Agilent Cat# 200523) according to the manufacturer's instructions.

P0WT-HA plasmid was used as a template. The following primers were used: Fw 5'-GGTTTTTGACATCACATGTGAACATGCCGTTGTCACTGTAGTCTAG-3'; Rv 5'-CTA-GACTACAGTGACAACGGCATGTTCACATGTGATGTCAAAAACC-3'.

The P0T124M-HA sequence was confirmed by automated sequencing analysis.

## Behavioral tests

**Accelerating rotarod analysis.** Only female N3 C57BL/6N congenics at the ages indicated in the text were used. All mice were tested in a session of three trials per day for five consecutive days. Rotarod conditions were set to an acceleration of 5 rotation per minute$^2$ (rpm$^2$), starting at a minimum velocity of 4 rpm and accelerating to a maximum velocity of 40 rpm. Each trial consisted of one acclimating run that was not scored. The next three runs were recorded and averaged. Each run was stopped when the mouse fell or passed completely underneath the rod (180° of rotation) consecutively twice. Motor performance were determined from 5 mice per genotype at 2 months of age; 8 WT, 11 $Mpz^{T124M/+}$ and 9 $Mpz^{T124M/T124M}$ at 6 months of cage; 4 WT, 10 $Mpz^{T124M/+}$ and 7 $Mpz^{T124M/T124M}$ at 12 months of ages.

**Beam walking test analysis.** Both male and female N5 C57BL/6N congenics at 6 months of age were used. The beam apparatus (OpenScience Russia, Cat# TS0806-M) consists of 80-cm long beams with a flat surface 12- or 6-mm wide resting 50 cm above the tabletop on two poles with a slope of 15%. A black box is placed at the end of the beam as a finish point. A lamp, with a 60-watt light bulb, shines light above the start point and serves as an aversive stimulus. Thirty minutes before training/testing, the mice were transported to the room containing the beam apparatus. On training days (2 days), each mouse crossed the 12-mm-wide beam three times and then the 6-mm-wide beam three times (10-min rest between training sessions on the two beams). On the testing day, each mouse crossed the 6-mm beam four times, and each trial was video recorded. The time to cross the beam and the number of slips were measured and averaged. 6 mice per genotype were analyzed.

## Electrophysiological analysis

Electrophysiological analyses were performed as described by Weinstock *et al* [89]. Mice were anesthetized, at the ages indicated in the text, with 2,2,2 tribromoethanol (Avertin; Sigma-Aldrich), 20 mg/mL in H$_2$O, and placed under a heating lamp. The sciatic nerve conduction velocity was obtained with steel monopolar needle electrodes. One pair of stimulating electrodes was inserted subcutaneously near the nerve at the ankle, a second pair of electrodes was placed at the sciatic notch, and a third pair was placed over the dorsum of the spine. The compound motor action potential was recorded with an active electrode inserted into the muscles in the middle of the paw and a reference needle in the skin between the first and second digits. Electrophysiological parameters were determined from: 8 WT, 8 $Mpz^{T124M/+}$ and 6 $Mpz^{T124M/T124M}$ sciatic nerves (11 mice total) at 2 months of age, 14 WT, 18 $Mpz^{T124M/+}$ and 20 $Mpz^{T124M/T124M}$ sciatic nerves (total 26 mice) at 6 months of age and 8 WT, 12 $Mpz^{T124M/+}$ and 14 $Mpz^{T124M/T124M}$ sciatic nerves (total 17 mice) at 12 months of age.

## Simoa plasma Nfl measurment

Mice, at the ages indicated in the text, were anesthetized with 2,2,2 tribromoethanol (Avertin; Sigma-Aldrich), 20 mg/mL in H$_2$O. Five hundred microliters of blood per mouse was taken by cardiac puncture and placed in tubed coated with EDTA. Four microliters of EDTA was added to each sample and gently mixed. Samples were incubated for 15 min on ice and centrifugated for 10 min at 3,500 rpm at 4°C. Supernatants (plasma) were transferred to new tubes and stored at -80°C until analysis. Plasma sample Nfl concentration was determined using the

in-house Simoa Nfl assay as described by Rohrer JD *et al* [90]. 9 mice per genotype were used at 2 months of age; 6 mice per genotype at 12 months of age.

## Morphological assessments

Mice at the ages indicated in the text were euthanized, and sciatic nerves and toes (digital nerves) were dissected, fixed in 2% glutaraldehyde, and stored at 4˚C until processing. For lumbar spinal cord sections, mice were anesthetized and perfused, as describe below with 2.5% glutaraldehyde and 4% paraformaldehyde. Nerves and spinal cords were then washed in phosphate buffer (0.12 M, pH 7.4), postfixed in 1% osmium tetroxide, dehydrated in increasing ethanol concentrations (50%, 70%, 90%, and 100%), incubated in propylene oxide, and finally embedded in Epon 100%. Nerves and spinal cords were then cut into semithin sections of 1-µm thickness or ultrathin sections of 80–85-nm thickness. Semithin sections were stained with toluidine blue 2% (phosphate buffer [0.12 M pH 7.4]). Ultrathin sections were stained with uranyl acetate (in dH$_2$O) and lead citrate. For semithin sections, images were acquired with the 100× lens objective and stitched using PTGui software v.10 (New House Internet Services BV) to reconstruct a complete image of the nerve. Morphological parameters (degenerative figures, SLI and onion bulbs) were then evaluated for the full nerve. Electron micrographs were used for quantifications of g-ratios (at least 100 myelinated axons per animal), axonal size (at least 300 myelinated axons per animal), and myelin periodicity (10 myelin sheaths per animal) as described by Belin *et al* [91]. Schwann cells were identified by the presence of a basal lamina, whereas macrophages were identified by characteristic morphologic features, including microvilli. Quantifications were performed using ImageJ Fiji v1.52p. Between 3 and 6 mice per genotype per time point were analyzed.

For 3D EM reconstruction, tissue preparation and imaging were performed as previously described by Yin Xi *et al* [85]. In brief, mice at 12 months of age were perfused with 0.1-M sodium cacodylate buffer containing 2.5% glutaraldehyde (Electron Microscopy Sciences) and 4% paraformaldehyde. Sciatic nerves were dissected, cut into 5-mm segments, postfixed in 0.1% tannic acid in buffer and then stained successively with osmium ferricyanide, tetracarbohydrazide, aqueous osmium tetroxide, saturated aqueous uranyl acetate, and Walton's lead aspartate stains. Tissues were then dehydrated in graded ethanols and embedded in embedding resin at 60˚C for 48 h. Nerve segments were mounted on aluminum pins, trimmed, surrounded with silver paste, and examined in a Sigma VP scanning electron microscope (ZEISS) fitted with a 3View in-chamber ultramicrotome (Gatan) and a low-kilovolt backscattered electron detector (Gatan). Longitudinally oriented axons were imaged covering areas ∼80 µm wide and 80–400 µm in length. Images were collected at 1.8–2.5 kV, depending on tissue contrast, and up to 500 slices were cut at a thickness of 40–100 nm. Imaging was conducted using a 30-µm aperture in high-current mode and imaged at room temperature (21˚C at a chamber vacuum of $10^{-6}$ mbar and working distance of 5.7 mm). Images were reconstructed and registered using ImageJ/FIJI software (National Institutes of Health).

## Trans-cardiac perfusion

Mice were anesthetized with 20 mg/mL 2,2,2 tribromoethanol (Avertin; Sigma-Aldrich) (0.02 mL/g of body weight). Once the mouse was unconscious, the thoracic wall was removed and the right atria was punctured. Twenty-five milliliters of 1× PBS was perfused into the left ventricle, followed immediately by 25 mL of 4% paraformaldehyde. The spinal cord was dissected and postfixed in 4% paraformaldehyde for 24 h at 4˚C, followed by sucrose and OCT embedding. Tissues were frozen in OCT and stored at -80˚C.

## Teased fiber preparation

Slides were coated with 3-aminopropyl-triethoxysilane (TESPA; Sigma-Aldrich) by subsequently submerging glass slides in acetone for 1 min, 4% TESPA in acetone for 2 min, and two times in 4% TESPA in acetone for 30 s each. Nerves were fixed in 4% paraformaldehyde for 30 min and then washed with 1× PBS. Nerves were desheathed with forceps and a 27-gauge needle. Bundles of fibers were separated in 1× PBS using 27-gauge needles and then gently teased apart to single fibers on TESPA-coated slides using modified insulin syringes containing minutien pins (Fine Science Tools #26002–10) attached to their needles. Slides were allowed to dry for at least 1 h and stored at -80°C. Sciatic nerves from 3 to 5 mice per genotype per time point were teased and analyzed.

## Immunofluorescence

Teased fibers and spinal cord (20 μm), and sciatic nerve (10 μm) cross sections were rehydrated in 1× PBS for 1 min and permeabilized with cold methanol. Tissues were rinsed in 1× PBS three times for 5 min and blocked with 5% fish skin gelatin and 0.5% Triton X-100 (teased fibers) or with 20% fetal bovine serum, 2% bovine serum albumin (BSA), and 0.1% Triton X-100 (cross sections) for 1 h at room temperature. The following primary antibodies were diluted in appropriate blocking buffer for incubation overnight: 1:300 chicken anti-P0 (Aves Lab Cat# PZO), 1:200 mouse anti-KDEL (Enzo Life Sciences Cat# ADI-SPA-827), 1:1,000 rabbit anti-Caspr (a gift from Dr. Elior Peles [92]), 1:500 chicken anti-pan-Neurofascin (R&D Systems Cat #AF3235), 1:200 rabbit anti-Kv1.1 (Alomone Cat# APC-009), 1:500 rabbit anti-HSPD1 (also known as HSP60) (Proteintech Cat# 15282-1-AP), 1:500 chicken anti-Neurofilament M (BioLegend Cat# 822701), 1:100 fluorescein phalloidin (Thermo Fisher Scientific Cat# F432), 1:500 rabbit anti-MAG (Invitrogen Cat#34–6200), 1:300 goat anti-choline acetyltransferase (Millipore Cat# AB144P), 1:2 mouse anti-Cx32 (7C6.C7) (a gift from Dr. Steven Scherer [93]), and 1:2,000 mouse anti-tubulin b3 (TuJ1) (Covance Cat# MMS-435P). After washing three times with 1× PBS, the following secondary antibodies were diluted in blocking buffer and applied to sections for 1 h at room temperature: 1:500 Alexa 488 donkey anti-rabbit IgG (Jackson ImmunoResearch Cat# 711-545-152), 1:500 rhodamine (TRITC) donkey anti-rabbit IgG (Jackson ImmunoResearch Cat# 711-025-152), 1:500 Cy3 donkey anti-mouse IgG (Jackson ImmunoResearch Cat# 715-165-150), 1:500 Cy3 donkey anti-chicken IgY (Jackson ImmunoResearch Cat# 703-165-155), 1:500 Alexa 488 donkey anti-chicken IgY (Jackson ImmunoResearch Cat# 703-545-155), and 1:500 Cy3-AffiniPure donkey anti-goat IgG antibody (Jackson ImmunoResearch Cat# 705-165-003). DAPI was added for 5 min at room temperature. Slides were then washed three times with 1× PBS and mounted in Vectashield mounting medium. Staining for Cx32 (including Kv1.1) was similar to the general protocol, but sciatic nerves were not fixed before teasing. For analysis of Thy1-YFP, sciatic nerves were dissected, fixed in 4% paraformaldehyde for 1 h, and washed with 1× PBS. After a 10-min incubation in 0.1% Triton X-100 in 1× PBS, nerves were washed with 1× PBS, and the perineurium was carefully removed using a 27-gauge needle. Finally, tissue was whole mounted for visualization. Images were acquired with a confocal Leica SP5II or a Zeiss (Oberkochen, Germany) Apotome microscope. Images were analyzed using ImageJ Fiji v1.52p. Quantification of mitochondrial size was performed as described by Della-Flora Nunes *et al* [94], using a single stack of the axonal plane. Corrected total cell fluorescence was calculated as the integrated density–(area of selected cell × mean fluorescence of background readings).

## RNA isolation

Total RNA was isolated from mouse sciatic nerves using TRIzol (Life Technologies, Carlsbad, CA) reagent according to the manufacturer's instructions. Samples were frozen in liquid $N_2$

following dissection and stored at -80˚C. A cooled pestle was used to pulverize tissues to powder. The powder was then resuspended in TRIzol reagent. After incubation for 5 min on ice, chloroform was added and the mixture was shaken vigorously and centrifuged. The upper aqueous phase containing RNA was collected, and 1 μL glycogen was added (20 mg/mL). RNA was precipitated with isopropanol, pelleted, and washed twice with 75% ethanol. Pellets were resuspended in 10 μL DEPC-d$H_2O$. RNA was quantified (optical density at 260 nm) using a spectrophotometer (NanoDrop 2000C; Thermo Fisher Scientific) and analyzed for purity according to the 260/280 ratio.

## cDNA preparation and TaqMan qRT-PCR

An Invitrogen kit (Superscript III) was used to convert 1 μg of total RNA to cDNA using the oligo(dT) and hexamers provided. Following the reverse transcription reaction, the provided RNase H was added to each sample. cDNA was collected and stored at -80˚C. For RT-qPCR, TaqMan systems were used for different primers. The RT-qPCR reaction was 50˚C for 3 min, 95˚C for 10 min, 95˚C for 15 s, and 60˚C for 1 min (40 cycles). The amount of cDNA used in qRT-PCR reactions was determined by standard curves in accordance with Applied Biosystems protocols. Each cDNA sample was tested in triplicates for the presence of each gene of interest and for the standard (*GAPDH and 18s*) on the same plates. Target and reference gene PCR amplification was performed with Assays-on-Demand (Applied Biosystems Instruments): *GAPDH* (Mm99999915_g1), *Ddit3/Chop* (Mm00492097_m1), *XBP1s* (Mm03464496_m1) and *Hspa5/BIP* (Hs99999174_m1), *18S* (Hs99999901_s1), *Mpz* (Mm00485141_g1), *Mbp* (Mm01266402_m1), *Id2* (Mm007011781_m), *c-Jun* (Mm00495062_s1), *Sox2* (Mm00488369_m1), *POU3F1/Oct6* (Mm00843534_m1), *Krox20/ Egr2* (Mm00456650_m1).

All samples were analyzed in triplicates, and the relative expression of the target RNAs was calculated using the $\Delta\Delta C_T$ of the gene of interest compared to the housekeeper gene. Reactions without target cDNA were used as a negative control for each reaction. Sciatic nerves from 3 to 6 mice per genotype per time point were analyzed.

## Protein extraction and Western blotting

Tissue or cells were lysed in RIPA buffer supplemented with phosphatase and protease inhibitors. For sciatic nerves, the samples were flash frozen in liquid $N_2$ and pulverized with a pestle. The crushed powder was resuspended in lysis buffer. For cells, wells were washed with 1× PBS, and RIPA buffer was added to each well on ice. Cells were scraped off in RIPA buffer and collected. Lysates were left on ice for 20 min and centrifuged at 13,000 × *g* for 20 min at 4˚C. Supernatant protein concentrations were determined with a BCA protein assay kit (Thermo Fisher Scientific) according to the manufacturer's instructions. For deglycosylation experiments, samples were treated with endoglycosidase H (New England BioLabs Cat# P0702) and PNGase F (New England BioLabs Cat# P0704) according to the manufacturer's instructions. Samples were prepared with 4× Laemmli buffer and lysis buffer. Five micrograms of protein was loaded per lane and resolved using SDS-polyacrylamide gel electrophoresis (SDS-PAGE) under denaturing conditions with a mini-Protean II gel electrophoresis apparatus; Precision Plus Standard Protein Dual color (Bio-Rad) was included to enable band size identification. Separated proteins were transferred to a polyvinylidene difluoride blotting membrane in a mini gel transfer tank. Blots were then blocked with 5% BSA in TBS-0.05% Tween 20 for 1 h at room temperature. The blots were incubated overnight at 4˚C with the following primary antibodies in 3% BSA in TBS-0.05% Tween 20: 1:5,000 chicken anti-P0 (Aves Lab Cat# PZO), 1:1,000 rat anti-HA high affinity (Roche Cat# 11867423001), 1:3,000 rabbit anti b-tubulin

(Novus Cat# NB600-936), 1:1,000 rabbit anti-PMP22 (Sigma Cat# SAB4502217), 1:1,000 rabbit anti-CNPase (Cell Signaling Cat# 5664), 1:1,000 rabbit anti-MAG (Invitrogen Cat# 34–6200), and 1:1,000 rabbit anti-HSP90a (Thermo Fisher Scientific Cat# PA3-013). Membranes were washed in TBS-0.05% Tween 20 three times for 5 min and incubated for 1 h at room temperature with the following horseradish peroxidase-conjugated secondary antibodies: 1:20,000 donkey anti-rabbit IgG(H+L) polyclonal (Novus Cat# NB7185) and peroxidase-AffiniPure donkey anti-chicken IgY antibody (Jackson ImmunoResearch Cat# 703–035–155). Blots were developed using ECL (GE Healthcare, Chicago, IL) and quantified using Image J software. Sciatic nerves from 5 mice per genotype per time point were analyzed.

## Metabolite extraction and measurement

Tissue metabolites extraction and measurement were performed as described by Sasaki *et al* [61]. Tissues were collected and immediately frozen in liquid nitrogen and stored at −80°C. Frozen tissues were homogenized by sonication (Branson Sonifier 450, output 2.5, 50% duty cycle, 10–20 s) in 50% MeOH in water (160 μl). Homogenates were centrifuged (13,000 $g$, 10 min, 4°C) and cleared supernatants were transferred to new tubes. One third volume of chloroform was added to the supernatant, mixed, and centrifuged (13,000 $g$, 10 min, 4°C). The aqueous phase (140 μl) was transferred to a new tube and lyophilized and stored at −20°C until analysis. Lyophilized samples were reconstituted with 5 mM ammonium formate (70 μl for the sciatic nerve), centrifuged (13,000 $g$, 10 min, 4°C). The 10 μl clear supernatant was mixed with 10 μl 5mM ammonium formate and loaded on LC-MS.

NAD+, NADH, ATP, ADPR, and NAM were measured using LC-MS/MS using C18 reverse phase column (Atlantis T3, 2.1 × 150 mm, 3 μm; Waters) equipped with HPLC system (Agilent 1290 LC) at a flow rate of 0.15 ml/min with 5 mM ammonium formate for mobile phase A and 100% methanol for mobile phase B. Metabolites were eluted with gradients of 0–10 min, 0–70% B; 10–15 min, 70% B; 16–20 min, 0% B. The metabolites were detected with a triple quadrupole mass spectrometer (Agilent 6470 MassHunter; Agilent) under positive ESI and multiple reaction monitoring (MRM) mode. Lactate, pyruvate, and G6P were measured using Metabolomics dMRM Database and Method (Agilent) according to the instructions. Serial dilutions of standards for each metabolite in 5 mM ammonium formate were used for calibration of NAD+, NADH, ATP, ADPR, and NAM. Metabolites were quantified by MassHunter quantitative analysis tool (Agilent) with standard curves (NAD+, NADH, ATP, ADPR, and NAM) or area under curves (lactate, pyruvate, and G6P) and normalized by the protein amount measured by BCA protein assay kit (Pierce).

## Auditory brainstem response (ABR)

The ABR was recorded in a sound attenuating chamber using a commercial system (SmartEP, Intelligent Hearing Systems, Miami, FL). Mice were anesthetized with ketamine (50 mg/kg, i. p.) and xylazine (6 mg/kg), placed on a temperature-controlled heating pad, and needle electrodes placed on the vertex (non-inverting), behind the ipsilateral pinna (inverting electrode) and the behind the contralateral pinna (ground). Neural responses were amplified, filtered (30–3000 Hz) and digitized (1024 presentations, 40 kHz sampling rate) in response to the click stimuli (1 ms rise/fall, cosine gated, 5 ms duration, 21/s) over a 10 ms window. The ABR was elicited with click stimuli to obtain well-defined peaks of wave I to V from which to estimate the latency of wave I. Click intensity was decreased from 90 to 30 dB pSPL in 10 dB steps). A wave I latency versus intensity function was generated for each WT (n = 5) and $Mpz^{T124M/T124M}$ (n = 5) mouse and the data used to construct a mean wave I latency-intensity function for WT and $Mpz^{T124M/T124M}$ mice.

## Cochleograms

Cytocochleograms were prepared as described previously [95,96]. Mice were killed with an overdose of $CO_2$, decapitated, the temporal bones quickly removed, the round and oval windows opened, and 10% formalin in phosphate buffered saline perfused into the cochlea. The cochleae were immersed in 10% formalin for 24 h, decalcified with 10% EDTA and stained with Harris' hematoxylin solution. The cochlear basilar membrane was dissected out, mounted as a flat surface preparation in glycerin on a glass slide, examined with a light microscope (400X) and the numbers of missing inner hair cells (IHCs) and outer hair cells (OHCs) counted along the entire length of the cochlea. Cochleograms were prepared showing the percentages of missing OHCs and IHCs as a function of percent distance from the apex were generated for each animal. Mean cochleograms were prepared for WT (n = 4) and $Mpz^{T124M/T124M}$ (n = 5) mice.

## Statistical analyses

Experiments were not randomized, but data collection and analysis were performed blindly to the conditions of the experiments. No statistical methods were used to predetermine sample sizes, but the sample sizes are similar to those generally employed in the field. All statistical analyses were conducted using GraphPad Prism 9.01. To determine the significance between genotypes, Student's *t* tests, one-way ANOVA, two-way ANOVA and Nested one-way ANOVA with Tukey's comparison tests were used. A *p* value of ≤0.05 was considered statistically significant.

## Supporting information

**S1 Fig. $Mpz^{T124M}$ mouse construction and sequence validation.** (**A**) The targeting construct to introduce the T124M mutation into exon 3 of the mouse *Mpz* gene is based on the construct used to engineer the R98C mouse. The neomycin resistance gene was removed by crossing the founder mice with CMV-Cre mice. (**B and C**) P0 mRNA from wild-type (WT) and $Mpz^{T124M/T124M}$ (T124M) sciatic nerves was cloned and sequenced. Nucleotide (**B**) and amino acid (**C**) sequences (P0 exon 3) for the wild type (WT) and mutant (T124M) were compared. T124M substitution is highlighted in purple. Note how close the T124M mutation is to the *N*-glycosylation acceptor site N122 (highlighted in blue). Strain-specific neutral mutation is indicated in green. More details are provided in **S1 Data**. (**D**) $Mpz^{T124M}$ mouse genotyping strategy. Introduction of T124M mutation generates a restriction recognition site for NSP1. This new site is located 8 bp from the 5′ end of a preexisting NSP1 restriction site. After amplification, DNA from the wild type (WT) is cut once by NSP1 to generate two fragments of 114 and 55 bp. DNA from the $Mpz^{T124M}$ mutant is cut at two different sites by NSP1 to generate three fragments of 106, 8, and 55 bp. (**E**) Representative genotyping PCR of WT, $Mpz^{T124M/+}$, and $Mpz^{T124M/T124M}$. WT has a band at 114 bp. $Mpz^{T124M}$ heterozygote exhibits the 114 bp band and the 106-bp mutant band. $Mpz^{T124M}$ homozygote shows only one band at 106 bp. The three genotypes have a band at 55 bp; 8 bp is too small to be detected.
(TIF)

**S2 Fig. Semithin transections of sciatic nerves reveal axonal degeneration in $Mpz^{T124M}$ mice.** Representative images of transverse semithin sections of sciatic nerves stained with toluidine blue from wild-type (WT) and $Mpz^{T124M/+}$ (TM/+) and $Mpz^{T124M/T124M}$ (TM/TM) mice at 2 (**A**), 6 (**B**), 12 (**C**), and 18 (**D**) months of age. Arrowheads indicate degenerative figures, arrows indicate regenerative axons, asterisks indicate Schmidt-Lanterman incisures (SLI), and

"ba" indicates a myelin balloon. Scale bars: 20 μm.
(TIF)

**S3 Fig. Myelin thickness in *Mpz*^T124M^ mice.** From 2 (**A**) to 6 (**C**) months of age, we observed similar myelin thickness between WT and *Mpz*^T124M^ nerves, whatever axon caliber. Relative frequency of g-ratio is equally distributed among WT and *Mpz*^T124M^ mutants at 2 (**B**) and 6 (**D**) months of age. g-ratio as a function of axonal diameter shows hypermyelination of small fibers in *Mpz*^T124M^ mice compared to that in WT mice at 12 (**E**) and 18 (**G**) months of age. Relative frequency of g-ratio is shifted toward smaller g-ratio values in *Mpz*^T124M^ fibers at 12 (**F**) and 18 (**H**) months of age.
(TIF)

**S4 Fig. Myelin proteins expression is not altered in *Mpz*^T124M^ mice.** Western blot analysis for wild-type (WT) and *Mpz*^T124M/+^ (TM/+) and *Mpz*^T124M/T124M^ (TM/TM) mice at 2 (**A**), 6 (**E**), and 18 (**I**) months of age. Blots were probed with CNP, PMP22, and P0 antibodies. β-Tubulin and HSP90α were used as loading controls. Densitometric quantification did not reveal an alteration of MAG (**B, F, and J**) or PMP22 (**D, H, and L**) expression. CNP expression was reduced in *Mpz*^T124M/T124M^ mice at 18 months of age (**K**) [F (2, 15) = 5.534, p = 0.0158] but not at younger ages (**C and G**). T124M mutation alters P0 migration. *n* (animals) ≥ 3 per genotype. *p < 0.05* by multiple-comparisons Tukey's *post hoc* tests after one-way ANOVA (**B to D**, **F to H**, and **J to L**). Graphs indicate means ± SEMs.
(TIF)

**S5 Fig. Expression of myelin genes and master myelination regulators transcripts.** Total RNA was extracted from wild-type (WT), *Mpz*^T124M/+^ (TM/+) and *Mpz*^T124M/T124M^ (TM/TM) sciatic nerves at 2 and 12 months of age. Quantitative real-time PCR experiments were performed using primers recognizing *Sox2* (**A and B**), *Id2* (**C and D**), *c-Jun* (**E and F**), *Krox20* (**G and H**), *Oct6* (**I and J**), *Mpz* (**K and L**) and *Mbp* (**M and N**). The qRT-PCR was normalized using *18S* RNA. *n* (animals) ≥ 3 per genotype. Multiple-comparisons Tukey's *post hoc* tests after one-way ANOVA. Graphs indicate means ± SEMs.
(TIF)

**S6 Fig. Semithin transections of digital nerves from toes reveal fiber degeneration in *Mpz*^T124M^ mice with an FVB/N background.** Representative images of transverse semithin sections of digital nerves stained with toluidine blue from two wild-type (WT) and two *Mpz*^T124M/T124M^ (TM/TM) mice at 12 months of age in FVB/N background. Scale bars: 20 μm.
(TIF)

**S7 Fig. P0T124M mutation does not alter Remak bundles.** (**A**) Representative electron micrograph of Remak bundles of wild-type (WT), *Mpz*^T124M/+^ (TM/+) and *Mpz*^T124M/T124M^ (TM/TM) sciatic nerves. Scale bar: 2μm. As in WT, in *Mpz*^T124M^ axons are uniformly ensheathed by SC cytoplasm. (**B**) Degenerating axons were not observed in *Mpz*^T124M^ mutants Remak bundles. (**C**) Quantification of Remak bundles area (μm²). (**D**) Quantification of Remak bundles axon density (axon per μm). (**E**) Quantification of axon diameters size (μm). *n* (animals) ≥ 3 per genotype; at least 390 axons per genotype were quantified. Multiple-comparisons Tukey's *post hoc* tests after one-way ANOVA (**B**) and Nested one-way ANOVA (**C, D, E**). Graphs indicate means ± SEMs.
(TIF)

**S8 Fig. ATF3 expression in *Mpz*^T124M^ spinal cord.** Representative confocal microscopy images of spinal cords sections (L3-L5) from 2 and 12-month-old wild-type (WT), *Mpz*^T124M/+^ (TM/+) and *Mpz*^T124M/T124M^ (TM/TM)–ATF3-GFP mice stained for TuJ1 (red) (**A**), choline

acetyltransferase (CHAT; motoneurons) (red) (**B**) and ATF3-GFP (green). Scale bars: 40 μm. High-magnification insets show motoneuron expressing GFP under ATF3 promoter control. (**C**) Representative images of transverse semithin sections of lumbar spinal cords stained with toluidine blue from WT, $Mpz^{T124M/+}$ and $Mpz^{T124M/T124M}$ mice at 12 months of age. $n$ (animals) $\geq$ 3 per genotype.
(TIF)

**S9 Fig. Schematic hypothetical P0T124M pathomechanism.** P0T124M mutation impedes *N*-glycosylation and is responsible for additional P0 modifications. Axon–glia exchange areas (paranodes, Schmidt-Lanterman incisures [SLI], and gap junctions) are altered in $Mpz^{T124M}$ mutants and could lead to deficient transport of metabolites from Schwann cells (SC) to axons. Lack of SC support deprives axons of ATP. Axonal transport is slowed, leading to mitochondrial fragmentation, aggregation, and degeneration. Damaged mitochondria are not able to produce ATP, inducing a vicious cycle. Mitochondrial stress could activate SARM1, which cleaves $NAD^+$. $NAD^+$ depletion leads to axonal degeneration.
(TIF)

**S1 Movie. Example of 6-month-old WT mouse performing beam walking test.** Related to Fig 1.
(MP4)

**S2 Movie. Example of 6-month-old $Mpz^{T124M/+}$ mouse performing beam walking test.** Related to Fig 1.
(MP4)

**S3 Movie. Example of 6-month-old $Mpz^{T124M/T124M}$ mouse performing beam walking test.** Related to Fig 1.
(MP4)

**S1 Data. Sequencing of $Mpz$ mRNA from WT and $Mpz^{T124M/T124M}$ sciatic nerves.**
(DOCX)

**S2 Data. All numerical for quantitation figures.**
(XLSX)

## Acknowledgments

We thank Dr. Caterina Berti and Courtney Williamson for their excellent technical assistance, Dr. Karen Dietz for assistance preparing this manuscript.

## Author Contributions

**Conceptualization:** Ghjuvan'Ghjacumu Shackleford, M. Laura Feltri, Lawrence Wrabetz.

**Formal analysis:** Ghjuvan'Ghjacumu Shackleford, Lawrence Wrabetz.

**Funding acquisition:** Ghjuvan'Ghjacumu Shackleford, Maurizio D'Antonio, Lawrence Wrabetz.

**Investigation:** Ghjuvan'Ghjacumu Shackleford, Leandro N. Marziali, Yo Sasaki, Anke Claessens, Cinzia Ferri, Nadav I. Weinstock, Alexander M. Rossor, Nicholas J. Silvestri, Emma R. Wilson, Edward Hurley, Grahame J. Kidd, Senthilvelan Manohar, Dalian Ding, Richard J. Salvi.

**Methodology:** Ghjuvan'Ghjacumu Shackleford, Lawrence Wrabetz.

**Project administration:** Lawrence Wrabetz.

**Resources:** Yo Sasaki, Alexander M. Rossor, Grahame J. Kidd, Richard J. Salvi, M. Laura Feltri, Maurizio D'Antonio, Lawrence Wrabetz.

**Supervision:** Ghjuvan'Ghjacumu Shackleford, M. Laura Feltri, Maurizio D'Antonio, Lawrence Wrabetz.

**Validation:** Ghjuvan'Ghjacumu Shackleford, M. Laura Feltri, Maurizio D'Antonio, Lawrence Wrabetz.

**Visualization:** Ghjuvan'Ghjacumu Shackleford, Maurizio D'Antonio, Lawrence Wrabetz.

**Writing – original draft:** Ghjuvan'Ghjacumu Shackleford, M. Laura Feltri, Maurizio D'Antonio.

**Writing – review & editing:** Ghjuvan'Ghjacumu Shackleford, Richard J. Salvi, M. Laura Feltri, Maurizio D'Antonio, Lawrence Wrabetz.

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
