## [Decision Letter · Decision Letter 0]

22 Jun 2022

Dear Dr Shackleford,

Thank you very much for submitting your Research Article entitled 'MPZ-T124M mouse model replicates human axonopathy and suggest alteration in axo-glia communication' to PLOS Genetics.

The manuscript was fully evaluated at the editorial level and by independent peer reviewers. The reviewers appreciated the attention to an important topic but identified some concerns that we ask you address in a revised manuscript

We therefore ask you to modify the manuscript according to the review recommendations. Your revisions should address the specific points made by each reviewer.

[LINK]

Yours sincerely,

Gregory A. Cox

Associate Editor

PLOS Genetics

Gregory Barsh

Editor-in-Chief

PLOS Genetics

Reviewer's Responses to Questions

**Comments to the Authors:**

Reviewer #1: The authors describe a mouse model of the neuropathy (CMT2J) resulting from the T124M MPZ mutation. They provide convincing evidence for a predominantly axonal phenotype, as observed in human CMT2J patients. The paper is well written. It provides novel, interesting insight into the spectrum of MPZ functions and adds to our understanding of pathomechanisms of axonal and demyelinating neuropathies in general.

The authors present a large body of sound data; still, in order to provide sufficient support for the hypotheses raised, a few points still need to be addressed in a revised version of the manuscript.

“P0T124M mutation perturbs axon-glia interactions”: This should be examined in more detail by EM, specifically by looking for abnormal Schwann cell-axon networks and related alterations.

“Pathogenesis of P0T124M does not involve the unfolded protein response”: Was there ultrastructural evidence of ER stress and/or altered autophagy in Schwann cells? If not, please state explicitly that this was absent.

„Axonal transport and mitochondrial disruption in T124M mice“: Mitochondria / axoplasmic reticulum contacts should be examined. In addition, alterations of the axonal cytoskeleton (intraaxonal filamentous aggregates, altered microtubules, etc.) are of interest.

Pease address unmyelinated nerve fiber pathology.

Spinal cord alpha motor neurons did not appear to be reduced in number. But did they should any cytological abnormalities?

Did you examine DRG neurons?

Reviewer #2: Mutations in the peripheral myelin protein-zero (MPZ) result in peripheral neuropathy (Charcot Marie Tooth Disease) but can manifest itself in a range of distinct clinical pictures depending on the exact mutation. The missense mutation MPZ(T124M) is associated with CMT2J a form of CMT that is characterised by severe axonopathy but with surprisingly little evidence of demyelination, dismyelination or hypomyelination. To understand the patho-mechanism of this form of CMT and that of other forms of CMT primarily characterised by an axonopathy, the authors have generated a mouse model in which the exact missense mutation T124M is engineered in the mouse MPZ gene. The paper describes the meticulous characterization of mice heterozygous and homozygous for this mutation and find that it largely recapitulates the human clinical manifestation of CMT2J. In general, and in concordance with other mouse models for axonal CMT, the phenotype in mice is much milder than that observed in human CMT2J patients. The authors suggest that the pathomechanism of this form of CMT2 targets axonal mitochondria, causing axonal transport defects and mitochondrial disfunction leading to axonal degeneration and axonal loss over time. In future, this mouse model provides an excellent opportunity to study the exact pathological mechanisms of CMT2J and is a valuable pre-clinical model in which to test novel therapies that aim to preserve axonal integrity and function.

The experiments are very diverse and well-executed and underpin the major tenets of this study.

Minor but important point:

The gene and protein nomenclature adopted in this paper is a mess and does not conform to international standards and is furthermore inconsistent. I urge the authors to review this and make changes. For example, the gene name for Connexin32 is sometimes written as Cx32 and in other places as Cxn32 (Cxn32 in Figure 7 but Cx32 in the text!). The MGI name for this gene is Gjb1. If the authors want to stick to Cx32, then be consistent with proteins in all capital and gene names in italics (MPZ protein, Mpz (mouse gene) or MPZ (human gene). The same goes for Kv1.1; in line 447 and 448 written like KV1.1 (first time I see it denoted like this).

The nomenclature for the T124M mutations is very colloquial. Why not adhere to accepted nomenclature? Instead of TM/+, write MPZT124M/+ (in italics and T124M in superscript: formatting lost in editorial manager). And so on….

Please review line 451: ‘In general, human myelin sheaths are thin,… Rephrase.

Reviewer #3: In this study, the authors have developed and characterized a mouse model that harbors a T124M mutation in the myelin protein zero (MPZ) with the intent of modelling CMT2J, a hereditary axonal neuropathy which is characterized by the same mutation in humans. The manuscript includes comprehensive phenotypical, electrophysiological and histological analyses of mutant mice and convincingly confirms that the T124M substitution in MPZ results in axonal neuropathy. This is important as among the many rodent models for genetically determined neuropathies, the T124M mutant closes up to the only very few ones that are known where a mutated myelin protein does not result in demyelination but primarily in progressive axonal degeneration.

As expected, no overt abnormalities in compact myelin could be observed in mutant mice. In contrast, areas of non-compact myelin appear largely disorganized, which could be explained by altered MPZ glycosylation and protein targeting. This finding is of particular interest as non-compact myelin domains are thought to constitute routes of metabolic support for axons. N-glycosylated MPZ could play a role in securing non-compact myelin integrity and hence axonal support. The authors could observe axonal degeneration and reduced levels of ATP and NAD+ in mutant mice which may hint to SARM1 mediated axonal degeneration downstream of impaired metabolic support by Schwann cells. As metabolic support by myelinating glia is one of the experimentally most challenging aspects in axo-glia interaction, no complete resolution of the malfunction at the axo-glial interface can be expected at this point and the primary axonal damage without demyelination is a finding that is striking enough in its own. However, the metabolite measurements were performed in nerve lysates and cannot unequivocally resolve the cellular source.

Overall, the manuscript is well written, the methodology employed is adequate and of high standard, and although widely descriptive in nature, the manuscript is of high relevance to the field. However, despite these appreciations, there are certain issues that need to be addressed.

1) The source of the measured metabolites, i.e. glial vs axonal, is not resolved. NAD+, for instance, may also play a role in Schwann cells (PMID: 29921717). This limitation should be discussed.

2) The statement “Glycolytic activity in myelinating glia is fundamental for axonal energetics and survival (45–50).”, is misleading and should clearly be confined to the CNS, as it is also correctly phrased in the discussion: “However, the lactate and glycolysis pathways play a limited role in axon survival in the PNS under physiological conditions (47,67–69).”.

3) The presence of ATF3-GFP in the DRG is interesting. Would the authors claim that this is the consequence of only affected myelinated sensory fibers? A comment on Remak bundle integrity (which according to the metabolite hypothesis of the authors should presumably be normal) would be helpful.

4) A quantification of Schwann cell numbers and at least rough molecular characterization of the Schwann cell differentiation/de-differentiation state would add insightful information to the manuscript.

5) It is apparent that the number of samples used to quantify glycolysis is far lesser than the ones used for other metabolites. Given the intrinsic variation observed, the authors cannot conclude definitively that glycolysis is not affected at all. As the authors prefer using parametric tests to assess statistical significance, the number of samples invariably affects the assessment of statistical significance.

6) There are some major concerns in how the authors choose to perform statistical testing in experiments involving repeated measures from the same biological replicate. For instance, for the G-Ratio analysis in Fig2, the authors compare only the mean G ratios of animals even though hundreds of axons contribute to the mean, which is correct. However, in Figures 5, 6, 7 and 8, the authors compare all the measures made from all the biological replicates en masse. This results in inflated P values and the analysis has very little statistical power. To standardize this issue, I strongly suggest that the authors represent the data as Superplots (PMID: 32346721) which represent both the measures of the technical replicates (multiple measures from the same sample) as well as the resultant mean measurement of the biological replicate. The authors can then perform a Nested ANOVA that factors both the mean and the technical replicates that contribute to the mean.

Minor:

1. The authors need to standardize all the data plots by showing the samples. Many bar plots do not contain sample information. Also, g-ratios reflect non-continious data and shouldn’t be plotted as bar charts.

2. While performing post tests for ANOVA, the authors need to standardize the use of Tukey’s or Dunnett’s. Whether they are interested in comparing the mutants to the WT animals or in comparing all the three groups needs to be standardized (See Fig1 vs all other Figures).

3. While representing the results of an ANOVA, the authors need to indicate the statistical significance of the ANOVA first, followed by the statistical significance of the posttest/multiple comparisons. If the ANOVA in itself does not show any statistical significance, then the posttests do not hold any statistical power by themselves. Please check the behavior ANOVA analysis provided in Fig 1.

4. The raw data needs to be properly annotated so that it can be consulted without hassle. Sample and other data features like time or individual measure before averaging is missing for a lot of data sets.

**Have all data underlying the figures and results presented in the manuscript been provided?**

Reviewer #1: Yes

Reviewer #2: Yes

Reviewer #3: Yes

PLOS authors have the option to publish the peer review history of their article (what does this mean?). If published, this will include your full peer review and any attached files.

Reviewer #1: No

Reviewer #2: No

Reviewer #3: **Yes: **Robert Fledrich

---

## [Decision Letter · Decision Letter 1]

13 Oct 2022

Dear Dr Shackleford,

We are pleased to inform you that your manuscript entitled "A new mouse model of Charcot-Marie-Tooth 2J neuropathy replicates human axonopathy and suggest alteration in axo-glia communication" has been editorially accepted for publication in PLOS Genetics. Congratulations!

Yours sincerely,

Gregory A. Cox

Academic Editor

PLOS Genetics

Gregory Barsh

Editor-in-Chief

PLOS Genetics

Comments from the reviewers (if applicable):

Reviewer's Responses to Questions

**Comments to the Authors:**

Reviewer #1: The points I raised have been addressed sufficiently in the revised version. In my opinion, the manuscript can now be accepted for publication.

Minor points, not requiring resubmission:

- Please address the axonal changes in two of the partially depicted myelinated fibers in the upper right and lower left corner of Fig. S7 TM/TM.

- Please check grammar and spelling in this new paragraph: “Ultrastructural image analysis of Remak bundles of MpzT124M showed that, as in WT, the axons were homogenously wrapped by SC cytoplasmic extensions. We did not noticed axonal abnormalities such as ploy-axonal pockets or degenerative axons in MpzT124M (35) (S7A and S7B Fig). Quantitative analysis demonstrated that Remak bundles size (S7C Fig) and axon density (S7D Fig) were consistent among genotypes. Finally, we measured Remak bundles associated axonal diameter. In MPZT124M mice, the mean axonal diameter and the proportion of large axons (1μm) were identical to WT, suggesting absence of axonal sorting defect (36). Because Remak bundles associated axons are unmyelinated and SC wrapping those axons do not express P0 protein, despite a lightly active P0 promoter (37), these 320 results suggest that axonal degeneration observed in MpzT124M mice is dependent of P0 protein expression and of the presence of myelin sheath.”

- Please replace „Remake bundles“ by “Remak bundles” (line 1365)

Reviewer #2: The authors have adequately addressed my concerns

Reviewer #3: The authors have responded to all remarks and added substantial new data which together significantly improved the manuscript. I have no further reservations and would like to congratulate and thank the authors for this meaningful study.

**Have all data underlying the figures and results presented in the manuscript been provided?**

Reviewer #1: Yes

Reviewer #2: Yes

Reviewer #3: Yes

PLOS authors have the option to publish the peer review history of their article (what does this mean?). If published, this will include your full peer review and any attached files.

Reviewer #1: No

Reviewer #2: No

Reviewer #3: **Yes: **Robert Fledrich

**Data Deposition**

http://datadryad.org/submit?journalID=pgenetics&manu=PGENETICS-D-22-00567R1

**Press Queries**

---

## [Editor Report · Acceptance letter]

4 Nov 2022

PGENETICS-D-22-00567R1 

A new mouse model of Charcot-Marie-Tooth 2J neuropathy replicates human axonopathy and suggest alteration in axo-glia communication 

Dear Dr Shackleford, 

We are pleased to inform you that your manuscript entitled "A new mouse model of Charcot-Marie-Tooth 2J neuropathy replicates human axonopathy and suggest alteration in axo-glia communication" has been formally accepted for publication in PLOS Genetics! Your manuscript is now with our production department and you will be notified of the publication date in due course.

With kind regards,

Anita Estes

PLOS Genetics

On behalf of:
